# Erythropoietin signaling regulates heme biosynthesis

Jacky Chung[1], Johannes G Wittig[1†], Alireza Ghamari[2], Manami Maeda[1‡], Tamara A Dailey[3,4], Hector Bergonia[5], Martin D Kafina[1], Emma E Coughlin[6], Catherine E Minogue[7], Alexander S Hebert[6], Liangtao Li[8], Jerry Kaplan[8], Harvey F Lodish[9], Daniel E Bauer[2,10], Stuart H Orkin[2,10], Alan B Cantor[2,10], Takahiro Maeda[1‡], John D Phillips[5], Joshua J Coon[6,7,11], David J Pagliarini[12], Harry A Dailey[3,4], Barry H Paw[1,2,10*]

[1]Division of Hematology, Brigham and Women's Hospital, Harvard Medical School, Boston, United States; [2]Division of Hematology-Oncology, Boston Children's Hospital, Harvard Medical School, Boston, United States; [3]Department of Microbiology, University of Georgia, Athens, United States; [4]Department of Biochemistry and Molecular Biology, University of Georgia, Athens, United States; [5]Division of Hematology and Hematologic Malignancies, University of Utah School of Medicine, Salt Lake City, United States; [6]Genome Center of Wisconsin, Madison, United States; [7]Department of Chemistry, University of Wisconsin-Madison, Madison, United States; [8]Department of Pathology, University of Utah School of Medicine, Salt Lake City, United States; [9]Whitehead Institute for Biomedical Research, Massachusetts Institute of Technology, Cambridge, United States; [10]Department of Pediatric Oncology, Dana-Farber Cancer Institute, Harvard Medical School, Boston, United States; [11]Department of Biomolecular Chemistry, University of Wisconsin-Madison, Madison, United States; [12]Department of Biochemistry, University of Wisconsin-Madison, Madison, United States

*For correspondence: bpaw@rics.bwh.harvard.edu

Present address: †School of Biological Sciences, University of East Anglia, Norwich, United Kingdom; ‡Center for Cellular and Molecular Medicine, Kyushu University Hospital, Fukuoka, Japan

Competing interests: The authors declare that no competing interests exist.

**Abstract** Heme is required for survival of all cells, and in most eukaryotes, is produced through a series of eight enzymatic reactions. Although heme production is critical for many cellular processes, how it is coupled to cellular differentiation is unknown. Here, using zebrafish, murine, and human models, we show that erythropoietin (EPO) signaling, together with the GATA1 transcriptional target, *AKAP10*, regulates heme biosynthesis during erythropoiesis at the outer mitochondrial membrane. This integrated pathway culminates with the direct phosphorylation of the crucial heme biosynthetic enzyme, ferrochelatase (FECH) by protein kinase A (PKA). Biochemical, pharmacological, and genetic inhibition of this signaling pathway result in a block in hemoglobin production and concomitant intracellular accumulation of protoporphyrin intermediates. Broadly, our results implicate aberrant PKA signaling in the pathogenesis of hematologic diseases. We propose a unifying model in which the erythroid transcriptional program works in concert with post-translational mechanisms to regulate heme metabolism during normal development.

## Introduction

Heme biosynthesis is a fundamental biological process that is highly conserved and involves eight enzymatic reactions that occur both in the cytosol and mitochondria (*Severance and Hamza, 2009*). In vertebrates, the most recognized role of heme is to serve as the oxygen-binding moiety in

**eLife digest** Heme is an iron-containing compound that is important for all living things, from bacteria to humans. Our red blood cells use heme to carry oxygen and deliver it throughout the body. The amount of heme that is produced must be tightly regulated. Too little or too much heme in a person's red blood cells can lead to blood-related diseases such as anemia and porphyria. Yet, while scientists knew the enzymes needed to make heme, they did not know how these enzymes were controlled.

Now, Chung et al. show that an important signaling molecule called erythropoietin controls how much heme is produced when red blood cells are made. The experiments used a combination of red blood cells from humans and mice as well as zebrafish, which are useful model organisms because their blood develops in a similar way to humans. When Chung et al. inhibited components of erythropoietin signaling, heme production was blocked too and the red blood cells could not work properly.

These new findings pave the way to look at human patients with blood-related disorders to determine if they have defects in the erythropoietin signaling cascade. In the future, this avenue of research might lead to better treatments for a variety of blood diseases in humans.

hemoglobin expressed by red blood cells (RBCs). During RBC maturation, heme metabolism genes are robustly upregulated (*Chung et al., 2012*; *Nilsson et al., 2009*; *Yien et al., 2014*). Not surprisingly, mutations in these genes are most commonly associated with hematologic defects in humans, underscoring the importance for a better understanding of the factors regulating heme biosynthesis. In particular, loss-of-function mutations in *FECH* (EC 4.99.1.1), which encodes the terminal rate-limiting enzyme in heme production, is strongly associated with the disease erythropoietic protoporphyria (EPP) (*Balwani and Desnick, 2012*; *Langendonk et al., 2015*).

The dependence of RBC biology on heme metabolism makes erythropoiesis an excellent system to gain insight into this process. Previous genetic analyses using RBCs have identified several mechanisms regulating heme metabolism most of which are transcriptional networks controlling mRNA expression of heme metabolism genes (*Amigo et al., 2011*; *Handschin et al., 2005*; *Kardon et al., 2015*; *Nilsson et al., 2009*; *Phillips and Kushner, 2005*; *Shah et al., 2012*; *Shaw et al., 2006*; *Wingert et al., 2005*; *Yien et al., 2014*). Currently, however, transcription-independent signaling mechanisms regulating heme production are poorly understood (*Chen et al., 2009*; *Paradkar et al., 2009*). Such mechanisms may play a critical role to couple heme metabolism to changes in the extracellular milieu, homeostasis, and development.

Here, we show that heme production is regulated by EPO/JAK2 signaling in concert with the GATA1 target, *Akap10* (*Fujiwara et al., 2009*). During red blood cell (RBC) development, PKA expression becomes increased at the mitochondrial outer membrane (OM) through AKAP10-dependent recruitment. We found that OM PKA catalytic (PKAc) subunits become disengaged from the autoinhibitory PKA regulatory (PKAr) subunits through direct interaction with phosphorylated STAT5 downstream of EPOR activation. Furthermore, we demonstrate that FECH is a kinase target of OM PKA and its phosphorylation triggers upregulation of its activity that is required to support erythropoiesis *in vivo*. Our work uncovers a previously unknown facet of heme metabolism with implications on human disease.

## Results

### Mitochondrial PKA expression increases with erythroid maturation

To begin examining post-translational mechanisms regulating heme metabolism, we performed an unbiased comparative analysis of the changing mitochondrial proteome in maturing RBCs. Mitochondria-enriched fractions isolated from undifferentiated and differentiated Friend murine erythroleukemia (MEL) cells were analyzed by quantitative mass spectrometry (*Pagliarini et al., 2008*) (*Figure 1A and B*). MEL cells have been reliably used to dissect the molecular mechanisms underlying hemoglobin production in erythroid cells (*Bauer et al., 2013*; *Canver et al., 2014*).

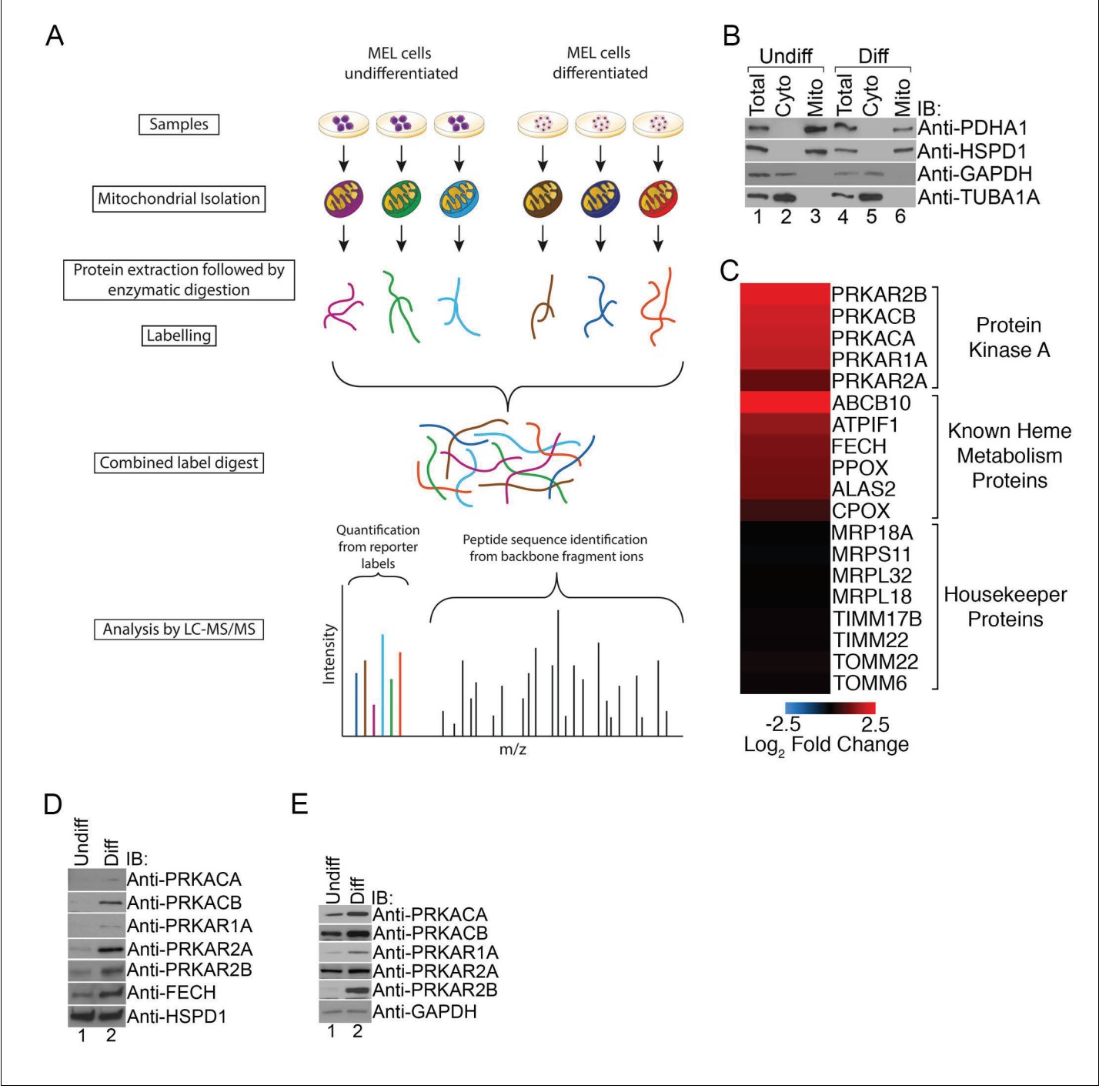

**Figure 1.** PKA activity regulates heme biosynthesis. (**A**) A schema detailing the preparation of samples enriched for mitochondria from undifferentiated and differentiated MEL cells is shown. Following preparation of samples from both cohorts, the samples were trypsin digested and labeled with different tandem mass tags (TMTs), followed by mass spectrometry analysis. (**B**) Enrichment of undifferentiated (undiff) and differentiated (diff) mitochondrial samples was confirmed by western analysis for mitochondrial (PDHA1 and HSPD1) and cytosolic (GAPDH and TUBA1A) markers prior to mass spectrometry analysis. (**C**) The upregulation of PKA regulatory (PKAr) and catalytic (PKAc) subunits as well as previously identified heme metabolism proteins in mitochondria-enriched fractions of differentiated MEL cells but not several housekeeping proteins is presented in logarithmic scale. Please see *Figure 1—source data 1* for precise changes. (**D and E**) Immunoblot analyses of the expression of PKA subunits in mitochondrial fractions (**D**) and whole cell lysates (**E**). All immunoblots were performed twice. Undiff-undifferentiated; Diff-differentiated; IB-immunoblot.

The following source data is available for figure 1:

*Figure 1 continued on next page*

*Figure 1 continued*

**Source data 1.** Changes in the mitochondrial expression of selection erythroid and housekeeping mitochondrial proteins.

As expected, erythroid differentiation was associated with the elevated mitochondrial expression of a number of mitochondrial proteins known to have a role in erythropoiesis such as FECH, ATPIF1, and ABCB10 while the expression of housekeeping proteins such as components of the mitochondrial transport and protein translation machinery were relatively unchanged (*Figure 1C*). Strikingly, we also found increased expression of PKA regulatory (PKAr; PRKAR1A, PRKAR2A, PRKAR2B) and catalytic (PKAc; PRKACA and PRKACB) subunits in mitochondria-enriched fractions (*Figure 1C*). We independently confirmed these results using immunoblotting with isoform-specific antibodies where we also found increased total expression of these PKA subunits in maturing erythroid cells (*Figure 1D and E*). We failed to detect PRKACG because it is only present in humans and not found in our murine model (*Kirschner et al., 2009*). In addition, we also could not detect PRKAR1B since its expression is restricted to neurons (*Kirschner et al., 2009*). Together, our results suggest that select PKA subunits become highly expressed in mitochondria of developing erythrocytes and that MEL cells are a good model that accurately recapitulates the expected PKA expression pattern.

## Mitochondrial PKA is localized to the outer mitochondrial membrane via AKAP10

We wondered whether increased mitochondrial PKA was specific for a particular suborganellular compartment, and next, performed a series of experiments to determine their submitochondrial expression. First, intact mitochondria isolated from maturing erythroid cells were treated with proteinase K that would digest all proteins exposed on the outer mitochondrial membrane. Immunoblot analysis of untreated and treated mitochondria revealed that the majority of PKA subunits were sensitive to proteinase K activity similar to TOM20 while VDAC1, a mitochondrial outer membrane (OM) marker known to be resistant to proteinase K digestion, remained largely unaffected (*Figure 2A*) (*Rapaport, 2003*; *Shirihai et al., 2000*). Biochemical fractionation of the mitochondria OM, intermembrane space (IMS), and mitoplast (MP) followed by immunoblotting confirmed the predominant presence of PKA subunits in the OM fraction of maturing erythroid cells (*Figure 2B*).

A great deal of work has demonstrated that PKA is localized to subcellular compartments through interactions with a family of anchoring proteins called AKAPs (a kinase anchoring proteins) (*Wong and Scott, 2004*). The majority of AKAPs recruit PKA-RII subunits but not RI (*Sarma et al., 2010*). However, a subclass of AKAPs can bind to both RI and RII with high affinity to regulate their subcellular distribution and have been referred to as 'dual-specificity AKAPs' (*Huang et al., 1997a*, *1997b*; *Li et al., 2001*; *Sarma et al., 2010*; *Wang et al., 2001*). In particular, PAP7, AKAP1, and AKAP10 are three such AKAPs capable of localizing to mitochondria (*Huang et al., 1997a*, *1997b*; *Li et al., 2001*; *Wang et al., 2001*; *Wong and Scott, 2004*). Interestingly, although we failed to detect PAP7 and AKAP1 in our proteomics analysis, we found a pronounced increase in the mitochondrial expression of AKAP10 (*Figure 2C and D*). Similar to our earlier data with PKA subunits, mitochondrial AKAP10 in maturing erythrocytes is sensitive to proteinase K digestion and primarily found in the OM fraction in maturing erythroid cells (*Figure 2E and F*).

High-throughput expression analysis has previously shown that *AKAP10* expression increases in maturing erythroid cells and is a downstream target of the GATA1 erythroid lineage master transcription factor (*Fujiwara et al., 2009*; *Zhang et al., 2003*). To date, it has no known role in erythropoiesis or heme metabolism. However, our results thus far led us to wonder if it was responsible for regulating PKA localization in maturing RBCs, and we tested this by using CRISPR/Cas9-mediated genome editing to introduce null mutations into the endogenous *AKAP10* loci. Genotyping and sequencing showed that for one *AKAP10* allele [$AKAP10^{Cas9(\triangle ex1-2)}$], parts of exons 1 (Ex1) and 3 (Ex3) along with all of exon 2 including the ATG start codon were deleted by our targeting strategy (*Figure 2G and H*). The second allele [$AKAP10^{Cas9(184delTG)}$] had a 2 base-pair deletion that resulted in a premature stop codon (Stop') (*Figure 2G and H*). Neither allele gave rise to full-length AKAP10 protein (KO) as shown by immunoblotting (*Figure 2H and I*).

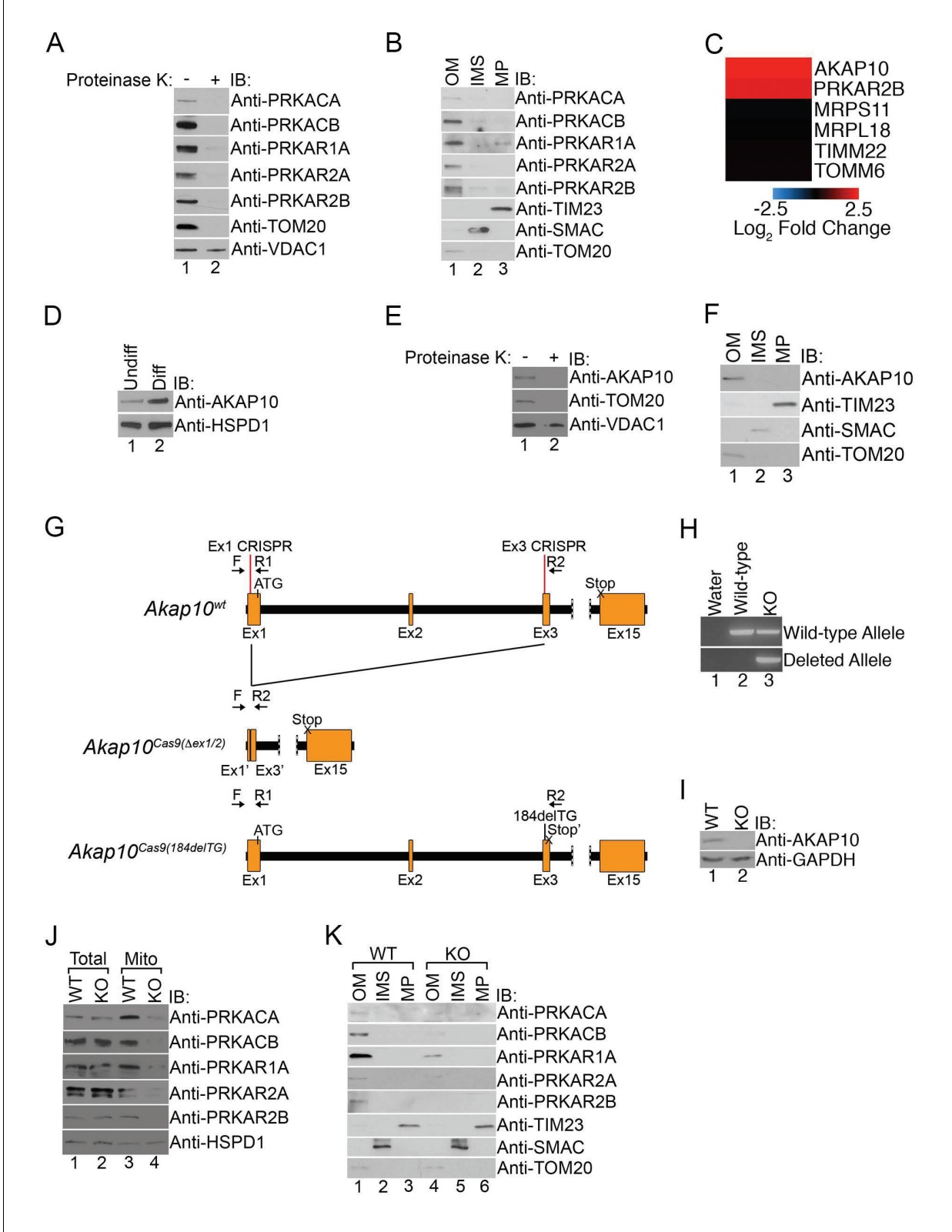

**Figure 2.** Mitochondrial PKA is localized to the outer membrane during erythropoiesis by AKAP10. (**A**) Intact mitochondria isolated from maturing MEL cells (day 3) were untreated or treated with proteinase K and subsequently analyzed by immunoblotted with antibodies specific for the indicated proteins. (**B**) Mitochondria from day 3 maturing MEL cells were fractionated into the indicated compartments and 5 μg of protein were analyzed with immunoblotting. (**C** and **D**) A heat map demonstrating the increased mitochondrial expression of AKAP10 similar to PRKAR2B in maturing erythroid

*Figure 2 continued on next page*

*Figure 2 continued*

cells (C) that was confirmed using immunoblotting (D). Please see *Figure 2—source data 1* for precise changes. (E and F) Proteinase K digestion assay (E) and submitochondrial fractionation (F) showed that AKAP10 is mostly localized to the OM. (G) A schematic depicting the wild-type (*Akap10^wt*) and the two *Akap10*-null alleles *Akap10^Cas9(△ex1-2/184delTG)* generated using CRISPR/Cas9 genome editing. The positions of the exon 1 (Ex1) and exon 3 (Ex3) CRISPR oligos are denoted. The introns are shown in black with exons in orange. The *Akap10^Cas9(ex1-2)* allele has complete removal of exon 2 and truncates exons 1 and 3 to fuse exons 1' and 3', respectively. The *Akap10^Cas9(184delTG)* allele has a two-nucleotide deletion in exon 3 leading to a frameshift and a premature stop codon (Stop'). Both alleles are expected to disrupt the N-terminal region encoding the mitochondrial-targeting motif. (H) The *Akap10^Cas9(△ex1-2)* deleted allele can only be detected when genotyping was performed with primers F and R2 while the *Akap10^Cas9(184delTG)* allele can still be detected with primers F and R1, resembling wild-type. These results were sequence confirmed. (I) Immunoblot analysis showing that neither allele gave rise to any detectable AKAP10 protein. (J and K) Loss of AKAP10 had no effect on total PKA subunit expression but reduced the amount of PKA subunits in whole mitochondria (J) as well as the OM fraction (K). All immunoblots were performed twice. Undiff-undifferentiated; Diff-differentiated; OM-outer membrane; IMS-intermembrane space; MP-mitoplast; WT-wild-type; KO-knockout; IB-immunoblot.

The following source data is available for figure 2:

**Source data 1.** Change in the mitochondrial expression of AKAP10 during erythroid maturation.

Total expression of PKA subunits in maturing KO cells was similar to wild-type (WT) cells (*Figure 2J*). However, KO cells had reduced the levels of mitochondrial PKA subunits both in intact preparations as well as in OM-specific fractions (*Figure 2J and K*). These results strongly suggest that AKAP10 recruits PKA to the outer mitochondrial membrane during red cell development and connects the GATA1 transcriptional program to the PKA signaling pathway.

## Mitochondrial outer membrane PKA signaling regulates hemoglobinization and erythropoiesis

Mitochondria are the site of heme production required for hemoglobin synthesis and its physiology is a crucial part of RBC maturation (*Nilsson et al., 2009*; *Shah et al., 2012*). Given the increase in mitochondrial expression of PKA subunits in maturing erythroid cells, we wondered whether PKA activity had an influence on heme production. We addressed this by first using pharmacologic agents to toggle PKA function. Compounds that activate PKA such as 8-Br-cAMP and forskolin both caused an increase in the proportion of hemoglobinized cells as shown by *o*-dianisidine staining, which can be blocked by PKA antagonists H-89 or PKI (14-22) (*Figure 3A*). An increase in the proportion of hemoglobinized cells was also observed when we treated MEL cells with dimethyl-prostaglandin E2 (dmPGE$_2$) (*Figure 3B*), which is a more stable analog of prostaglandin E2 (PGE$_2$) that has a physiologic role during multiple aspects of hematopoiesis (*Goessling et al., 2009*; *North et al., 2007*). The effects of dmPGE$_2$ can be similarly inhibited by PKI (14-22) (*Figure 3C*), underscoring the specificity of dmPGE$_2$ signaling via PKA. In contrast to PKA inhibition, the PKC inhibitor, *bis*-indolyl-maleimide II, could not block the effects of PKA activation suggesting that the observed changes in heme synthesis are specific for PKA (*Figure 3—figure supplement 1A*).

Although our results from pharmacologic experiments suggest that widespread modulation of PKA has an impact on heme production on maturing erythroid cells, they do not explicitly examine the role of mitochondrial OM PKA. The precise contributions of distinct mitochondrial pools of PKA have been a topic of controversy (*DiPilato et al., 2004*; *Lefkimmiatis et al., 2013*). Emerging evidence suggest that PKA agonists such as forskolin and cAMP cannot diffuse into the mitochondrial matrix (*Acin-Perez et al., 2009*; *Lefkimmiatis et al., 2013*). Thus far, our pharmacologic data involved the use of a high-dose of forskolin (*Figure 3A–C*), and when the dose was titrated down to one that was previously shown to not activate matrix PKA, we also failed to detect an effect on hemoglobinization (*Figure 3—figure supplement 1B*) (*Acin-Perez et al., 2009*).

It is difficult to rely solely on pharmacologic data to unambiguously dissect the contributions of intracellular PKA pools since dose responses are known to vary form one cell type to another (*Humphries et al., 2007*; *Lefkimmiatis et al., 2013*). However, the reduction in the levels of PKA subunits in the mitochondrial OM of AKAP10-KO maturing erythroid cells allowed us to genetically and biochemically examine the functional role of this PKA signaling compartment. Compared to wild-type controls, AKAP10-KO maturing erythroid cells exhibited a deficit in hemoglobinization

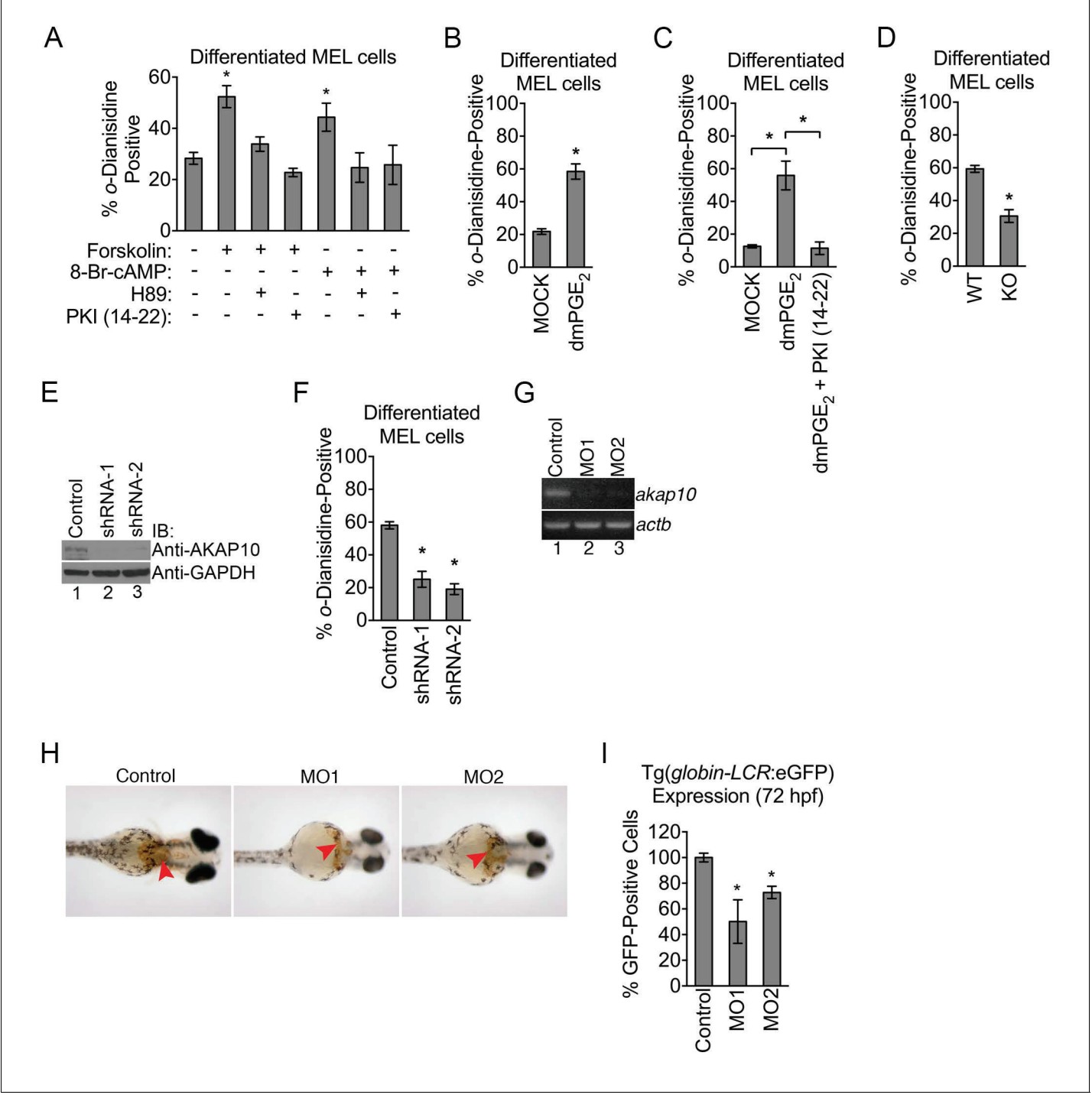

**Figure 3.** Mitochondrial outer membrane PKA signaling is required for erythropoiesis. (A–C) o-dianisidine staining for hemoglobinized MEL cells treated with several pharmacologic modulators of PKA activity. (A) PKA activation with 8-Br-cAMP or high-dose forskolin (50 μM) triggers an increase in heme production that is blocked by H89 or PKI (14-22) treatment at day 3 of DMSO induction. (B and C) A similar increase in hemoglobinization was observed with dMPGE$_2$ that was also inhibited by PKI (14-22). (D) Wild-type (WT) or AKAP10-knockout (KO) erythroid cells at day 4 of differentiation were stained with o-dianisidine. (E and F) AKAP10 expression was knocked-down using two different shRNAs (E) that lead to reduced hemoglobinization (F). (G–I) akap10-specific morpholinos (MOs) were used to inhibit akap10 expression in zebrafish embryos (G). These morphants were anemic with reduced hemoglobinization (H) and red blood cell counts (I). *p-value<0.05, Mean ± SEM, n = 3. All immunoblots were performed twice. 8-Br-cAMP-8-bromoadenosine 3′,5′-cyclic monophosphate; dmPGE$_2$-dimethyl-prostaglandin E$_2$; WT-wild-type; KO-knockout; shRNA-short hairpin RNA; MO-morpholino; IB-immunoblot.

*Figure 3 continued*

The following figure supplement is available for figure 3:

**Figure supplement 1.** PKA activity regulates heme biosynthesis.

(*Figure 3D*). This defect was also observed when AKAP10 expression was inhibited using two distinct shRNAs (*Figure 3E and F*).

We next examined the *in vivo* significance of AKAP10 and mitochondrial OM PKA signaling by using morpholinos to block *akap10* expression in zebrafish (*Danio rerio*) embryos (termed morphants) (*Figure 3G*). For over two decades, the zebrafish has been an invaluable model for the study of hematopoiesis and drug discovery (*Jing and Zon, 2011*; *Zon and Peterson, 2005*). Remarkably, *akap10* morphants were anemic with decreased hemoglobinization (*Figure 3H*, red arrowheads) compared to control embryos. We quantified these changes in red cell parameters by performing similar experiments on a transgenic zebrafish line in which all erythroid cells are marked by eGFP expression [Tg(*globin-LCR*:eGFP)] (*Ganis et al., 2012*). Flow-cytometry analysis revealed that *akap10* morphants had reduced RBC counts (*Figure 3I*). Together, our data suggest that mitochondrial OM PKA signaling is required for proper heme production and RBC development *in vivo*.

## The terminal heme enzyme, ferrochelatase, is directly phosphorylated by PKA

Next, we asked whether mitochondrial OM PKA signaling directly regulated heme biosynthesis by phosphorylating mitochondrial heme enzymes. Of the mitochondrial enzymes ALAS2, PPOX, CPOX, and FECH, the only enzyme with a predicted high-confidence PKA site (R/K-R/K-X-S/T-Z, where X is any amino acid and Z is an uncharged residue) is FECH at Thr116 (human FECH and Thr115 for murine FECH) (*Figure 4A*). This residue is evolutionarily conserved and is present on one of the lips of the active site pocket positioned in the middle of a long $\alpha$-helix (*Figure 4B–D*). In its unphosphorylated form, the side chain hydroxyl group of Thr116 (colored fuchsia) is sandwiched between His86 (colored blue) and Leu87 on an adjacent $\alpha$-helix, forming a hydrogen bond with His86 (*Figure 4C and D*) (*Wu et al., 2001*). Structural modeling suggests that the bulk of the added phosphate on the side chain of Thr116 would cause movement of the Thr116 $\alpha$-helix away from the His86 $\alpha$-helix, and thereby, shift the Thr116-containing $\alpha$-helix closer inwards towards the active site pocket opening and the porphyrin ring (shown in red) (*Figure 4C and D*). Such a modification may also destabilize the structure sufficiently to allow for more efficient movement of the active site lip during catalysis.

To test if FECH is directly targeted by PKA for phosphorylation, we performed an *in vitro* kinase assay by mixing together purified His-tagged human FECH and PKAc, followed by western analysis. This experiment showed that FECH is directly phosphorylated by PKA in an ATP-dependent fashion (*Figure 4E*). Using [$\gamma$-$^{32}$P]-ATP labeling, we calculated that, *in vitro*, 9.8 $\pm$ 3.2% of purified FECH was phosphorylated after 30 min. Substitution of Thr116 with Ala (T116A) abolished this phosphorylation (*Figure 4F*). In addition, consistent with the preference of PKAc for positively charged residues at the $-3$ and $-2$ positions (*Smith et al., 2011*), mutation of either Lys113 (K113L) or Arg114 to Leu (R114L) similarly reduced FECH phosphorylation (*Figure 4F*), strongly indicating that Thr116 of human FECH constitutes a *bona fide* PKA target. We also examined FECH phosphorylation in erythroid cells by performing similar immunoblot analysis. Immunoprecipitated FECH from differentiated MEL cells was detected by two different phospho-threonine antibodies—one targeting the Lys-X-X-pThr motif and another recognizing the pThr-Pro sequence (*Figure 4G*). High-dose forskolin treatment also increased phosphorylation of FECH in differentiating MEL cells (*Figure 4—figure supplement 1*). Conversely, inhibition of OM PKA with loss of AKAP10 resulted in diminished FECH Thr115 phosphorylation (*Figure 4H*). *In toto*, our results support a model where PKA becomes localized at the mitochondria OM of maturing erythroid cells and directly phosphorylates FECH.

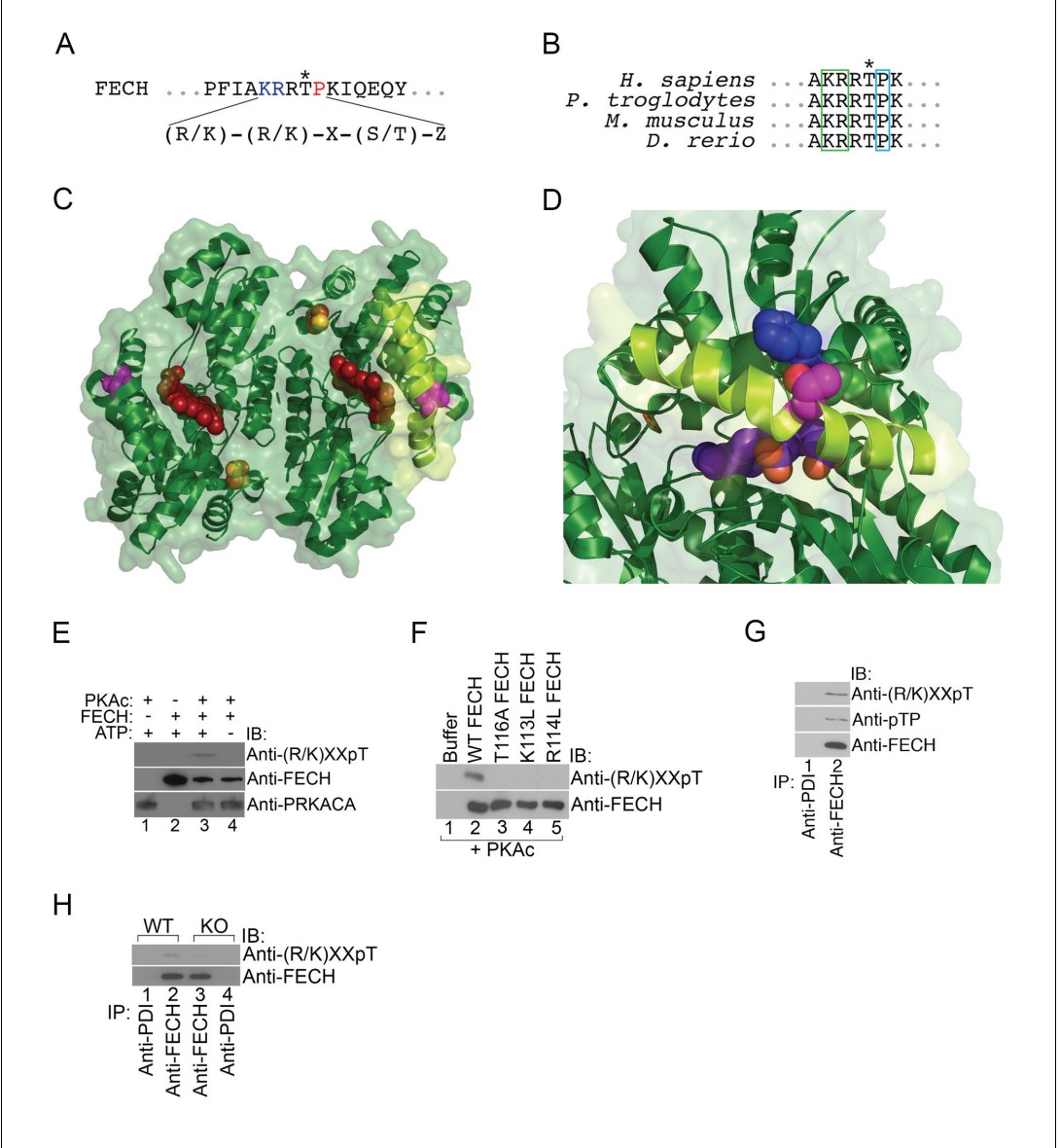

**Figure 4.** FECH is directly phosphorylated by PKA. (**A**) The motif surrounding Thr116 in human FECH constitutes a PKA phosphorylation site with a canonical Lys (K) and Arg (R) at positions −3 and −2, respectively, and an uncharged residue at the +1 position. (**B**) This PKA phosphorylation motif is highly conserved in FECH proteins across vertebrate species. (**C**) The FECH homodimer is shown with the transparent surface in green and the subunits in solid green ribbon. PPIX is shown as a red space filling model, the [2Fe-2S] clusters as solid rust and yellow balls, and the highlighted Thr116 (site of PKA-mediated phosphorylation) shown as solid violet spheres. The α-helix in which Thr116 resides, highlighted with lemon green, forms one lip of the opening to the active site where porphyrin is bound in this structure (PDB 2QD1). (**D**) The structure surrounding Thr116 (shown in fuchsia), which is situated in the middle of a long α-helix, is enlarged. Thr116 is in close proximity and hydrogen bonded to His86 (dark blue) and adjacent to Val85 (dark green). The Thr116-containing α-helix is highlighted with lemon green and the protoporphyrin behind in the active site is purple. Structural modeling suggests that phosphorylation of Thr116 would result in a shift of the Thr116-containing α-helix away from the His86-containing α-helix. (**E**) Purified recombinant His-tagged human FECH was phosphorylated by purified PKAc only in the presence of ATP at a 1:1 ratio. The phosphorylated form of FECH was detected by immunoblotting with an anti-phosphothreonine antibody. (**F**) An *in vitro* kinase assay was also performed with wild-type and variant forms of purified His-tagged human FECH. Disruption of the lysine or arginine at the −3 and −2 positions, respectively, similarly abolishes phosphorylation *in vitro* as a T116A mutation as shown by immunoblotting. (**G**) Differentiated MEL cells were immunoprecipitated with the indicated antibodies and immunoblotting analysis was performed. Immunoprecipitated FECH can be recognized by two different anti-phosphothreonine antibodies directed against the upstream positive residues and the proline immediately following the threonine. (**H**) Wild-type (WT) or AKAP10-KO (KO) were lysed and immunoprecipitated with the indicated antibodies. Bound proteins were analyzed by immunoblotting. All immunoblots were performed twice. ATP-adenosine triphosphate; PPIX-protoporphyrin IX; IB-immunoblot; IP-immunoprecipitate.

*Figure 4 continued on next page*

*Figure 4 continued*

The following figure supplement is available for figure 4:

**Figure supplement 1.** Activation of PKA with high-dose forskolin increases FECH phosphorylation.

## FECH phosphorylation is required for full activity during erythroid development

Past work has demonstrated that FECH is phosphorylated by protein kinase C (PKC) (*Sakaino et al., 2009*). PKC-mediated FECH phosphorylation occurs in a domain buried within an inaccessible hydrophobic fold that did not directly impact enzyme catalysis (*Sakaino et al., 2009*). In contrast, the position of Thr116 (Thr115 in mice) that is modified by PKA suggests that it would have a direct effect on FECH activity (*Wu et al., 2001*). We first examined this by using an *in vitro* $^{55}$Fe-based assay to measure and compare the amount of radiolabeled deuteroporphyrin-IX (DP) that can be produced by unmodified and modified FECH. DP, as a more soluble analog of the naturally occurring heme precursor protoporphyrin-IX (PPIX), is frequently employed in such measurements and is similarly metalated by FECH to generate deuteroheme (*Najahi-Missaoui and Dailey, 2005*). Significantly more radioactivity can be detected when purified His-tagged human FECH was added alone to the metalation reaction (*Figure 5A*) and the addition of purified PKAc to the reaction to catalyze the phosphorylation of FECH resulted in an approximately two-fold increase in $^{55}$Fe measurements (*Figure 5A*). This increase in activity is not attributable to PKAc per se, which has no ferrochelatase activity, and the reaction is completely DP-substrate dependent (*Figure 5A*). More detailed analysis on enzyme kinetics revealed that phosphorylation had a pronounced effect on maximum velocity ($v_{max}$) but did not significantly change the Michaelis-Menten ($K_m$) constant (*Figure 5B* and *Figure 5—figure supplement 1A*), suggesting that it has no major influence on substrate binding. An important caveat to these kinetic measurements is that they were performed at 25°C while all other in vitro assays were performed at 37°C and, thus, may not fully reflect enzyme kinetics both *in vivo* as well as other single time-point experiments. We also examined FECH activity in intact mitochondria isolated from maturing erythroid cells treated with a high dose of PKA-activating forskolin. Mitochondria from differentiating MEL cells exposed to high-dose forskolin catalyzed higher DP metalation compared to mitochondria derived from untreated cells (*Figure 5C*). In contrast, performing the assay with N-methyl mesoporphyrin-IX (NMMP)—a PPIX analog that is an inhibitor of FECH and not subject to metalation (*Dailey and Fleming, 1983*)—instead of DP, resulted is very low $^{55}$Fe extraction that was refractory to forskolin treatment (*Figure 5C*). Conversely, FECH activity was reduced in AKAP10-KO cells that had compromised FECH phosphorylation (*Figures 4H* and *5D*). These data indicate that phosphorylation of FECH at Thr116 by OM PKA increases FECH catalytic activity.

EPP patients harboring *FECH* mutations retain residual FECH activity (*Balwani and Desnick, 2012*), suggesting that subtle changes in FECH function have important biological implications. Thus, to examine the in vivo implications of FECH phosphorylation under physiological conditions, we used CRISPR/Cas9-directed homology repair to knock-in a T115A substitution into the endogenous *Fech* gene in murine erythroid cells (*Figure 5E*). Genotyping and subsequent sequencing confirmed that mutant cells possessed only the *Fech*$^{T115A}$ allele (*Figure 5F*). Compared to wild-type protein, FECH$^{T115A}$ mutant protein was similarly induced upon erythroid differentiation and was phosphorylation defective (*Figure 5G and H*). Direct measurements of enzyme activity from intact mitochondria isolated from wild-type and mutant maturing erythroid cells demonstrated that FECH$^{T115A}$ had diminished ferrochelatase activity (*Figure 5I*). Furthermore, *o*-dianisidine staining and high-performance liquid chromatography (HPLC) analysis revealed that erythroid cells expressing only FECH$^{T115A}$ protein had reduced hemoglobinization as well as lower intracellular hemin levels (*Figure 5J* and *Figure 5—figure supplement 1B*). Conversely, FECH$^{T115A}$-expressing cells had concomitantly elevated accumulation of the upstream, free protoporphyrin IX (PPIX) precursor (*Figure 5K*). Clinically, excess erythroid PPIX accumulation is only found in EPP cases where it serves as a diagnostic marker (*Balwani and Desnick, 2012*; *Whatley et al., 2004*). The build-up of PPIX in FECH$^{T115A}$-expressing cells strongly argues that this mutation specifically affects FECH function while the upstream heme biosynthetic pathway remains unaffected.

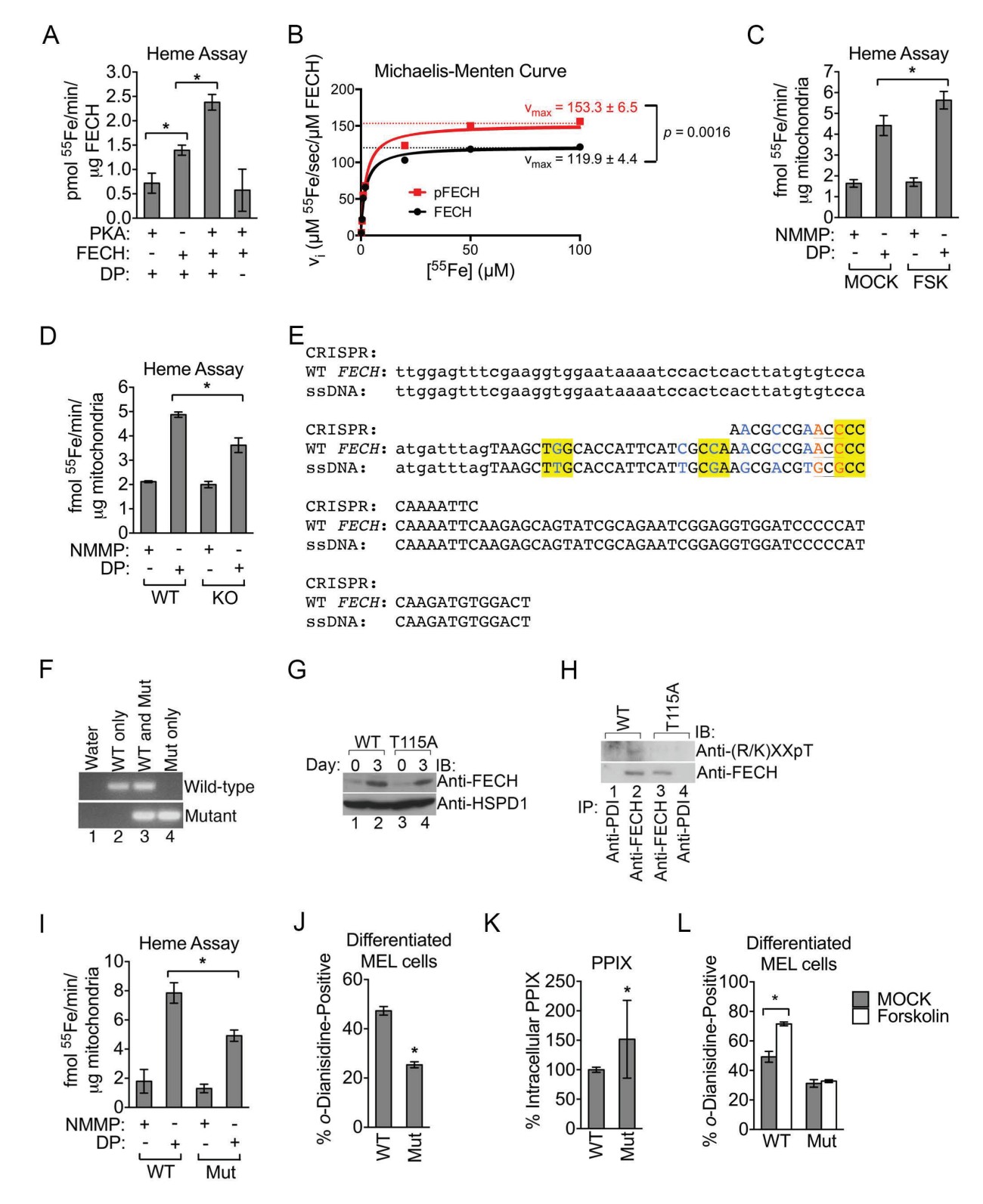

**Figure 5.** Phosphorylation of FECH is required for its full activity. (**A**) Single-point FECH activity was determined in vitro by measuring the amount of $^{55}$Fe incorporation into the protoporphyrin IX analog, deuteroporphyrin (DP). FECH has basal activity that is significantly increased by PKA-mediated phosphorylation. This increase in activity was dependent upon both DP and ATP, highlighting the substrate specificity of the assay for DP and the dependence on phosphorylation. (**B**) Kinetic analyses were subsequently performed with 0.1 μM FECH, 3 μM DP, and 0.2–100 μM $^{55}$FeCl$_3$ at 25°C.

*Figure 5 continued on next page*

*Figure 5 continued*

Phosphorylation of FECH leads to a statistically significant increase in maximum velocity ($v_{max}$). There was no significant difference in the $K_m$. Please see *Figure 5—source data 1* for $v_{max}$ and $K_m$ values. (C and D) FECH activity was measured in isolated intact mitochondria. Samples from high-dose forskolin (FSK)-treated differentiating MEL cells have higher FECH activity (C) (*p-value<0.05, Mean ± SEM, n = 5). In contrast, AKAP10-KO mitochondria had less FECH activity (D). Very little activity was detected in samples in which DP was substituted with NMMP. (E) A schematic showing the intron 3 and exon 4 sequences of wild-type murine *Fech* as well as the CRISPR oligo and the single-stranded DNA (ssDNA) that were introduced as a template for DNA repair. Intronic and exonic sequences are shown in lower and upper cases, respectively. Highlighted in yellow are the three PAM (protospacer adjacent motif) sequences closest to the T115A mutation site that facilitates the potential use of multiple CRISPR oligos. The missense mutations necessary to generate the T115A substitution are in orange. Shown in blue are synonymous substitutions designed to either disrupt the PAM sequences to prevent cleavage of the newly introduced mutant allele or to facilitate genotyping using allele-specific primers near the T115A mutation site. (F) Genomic DNA was isolated from the individual clones of MEL cells and used for PCR analysis with allele-specific primers. The parental MEL cells only had the wild-type allele. The intron 3 and exon 4 sequences of these cells were sequenced to confirm these agarose gel electrophoresis results. (G) Undifferentiated and differentiated parental and mutant cells expressing only the FECH[T115A] allele were lysed and subjected to western analysis to examine the induction of FECH protein during erythroid maturation. FECH[Thr115Ala] protein had very similar up regulation with differentiation. (H) Differentiated MEL cells were lysed, immunoprecipitated with the indicated antibodies, and bound proteins were subjected to western analysis. Cells expressing only mutant FECH were phosphorylation defective at Thr115. (I–K) Mitochondria isolated from differentiated MEL cells expressing only endogenous FECH[T115A] (Mut), generated by genome editing, has lower FECH activity than wild-type (WT) control (I). These cells expressing non-phosphorylated FECH also have reduced hemoglobinization by *o*-dianisidine staining (J) and increased accumulation of PPIX substrate (K) as demonstrated by HPLC analysis (*p-value<0.05, Mean ± SEM, n = 11). (L) Wild-type or mutant differentiated MEL cells treated with vehicle (MOCK) or FSK were stained with *o*-dianisidine. Cells expressing mutant FECH were refractory to the effects of FSK. *p-value<0.05, Mean ± SEM, n = 3, unless otherwise specified. All immunoblots were performed twice. IB-immunoblot; $v_i$-initial velocity; $K_m$-Michaelis-Menten constant; FSK-forskolin; NMMP:N-methyl-mesoporphyrin-IX.

The following source data and figure supplement are available for figure 5:

**Source data 1.** Maximum velocity ($v_{max}$) and Michaelis-Menten ($K_m$) constants.

**Figure supplement 1.** Phosphorylation is required for full FECH activity.

## Erythropoietin signaling activates PKA to phosphorylate FECH

During normal and stress erythropoiesis, erythropoietin (EPO) signaling through its cognate receptor tyrosine kinase (EPOR) regulates survival and proliferation of erythroid progenitors (*Kuhrt and Wojchowski, 2015*; *Testa, 2004*). However, there is evidence to suggest that EPO/EPOR signaling regulates other key aspects of erythropoiesis. Unfortunately, the study of such mechanisms has been hampered by the requirement of EPOR signaling in the early stages of the erythropoietic hierarchy (*Beale and Chen, 1983*; *Chida et al., 1999*; *Socolovsky et al., 1999*; *Testa, 2004*). Nevertheless, uncovering EPOR effectors in later differentiation stages has important clinical relevance. For example, limiting erythroid iron uptake dramatically ameliorates Polycythemia Vera (PV) symptoms in a murine model expressing constitutively active V617F JAK2 (*Ishikawa et al., 2015*). This is likely due to STAT5-mediated transcriptional up-regulation of the transferrin receptor (*Zhu et al., 2008*). Despite this, neither PV murine models nor EPO overexpressing transgenic mice show any signs of iron overload (*Li et al., 2011*; *Vogel et al., 2003*), indicating that EPO signaling not only promotes iron uptake in RBCs but also coordinates its physiologic assimilation into heme without excessive iron accumulation.

EPOR activates several pathways including the janus kinase 2 (JAK2)/signal transducer and activator of transcription 5 (STAT5), mitogen activated protein kinase (MAPK), and phosphatidyl 3'-inositol kinase (PI3K) pathways (*Kuhrt and Wojchowski, 2015*), leading us to question whether PKA is also activated by EPO. We tested this by treating primary murine fetal liver erythroblasts with EPO to determine if EPOR activation had any effect on PKA signaling by monitoring phosphorylation of the prototypical PKA target, CREB, on Ser133 (*Altarejos and Montminy, 2011*; *Taylor et al., 2012*). As expected, EPOR activation triggered Tyr694 STAT5 phosphorylation, which was similarly observed with EPO treatment of human UT7 erythroid cells (*Figure 6A–C*). CREB[Ser133] phosphorylation but not STAT5[Tyr694] phosphorylation was blocked by PKI (14-22) (*Figure 6D*), indicating that EPOR signaling specifically activates PKA. In contrast, co-treatment of UT7 cells with the JAK2 inhibitor, Ruxolitinib, robustly inhibited both STAT5[Tyr694] and CREB[Ser133] phosphorylation (*Figure 6E*), strongly suggesting that PKA lies downstream of the EPOR/JAK2 pathway. Co-immunoprecipitation

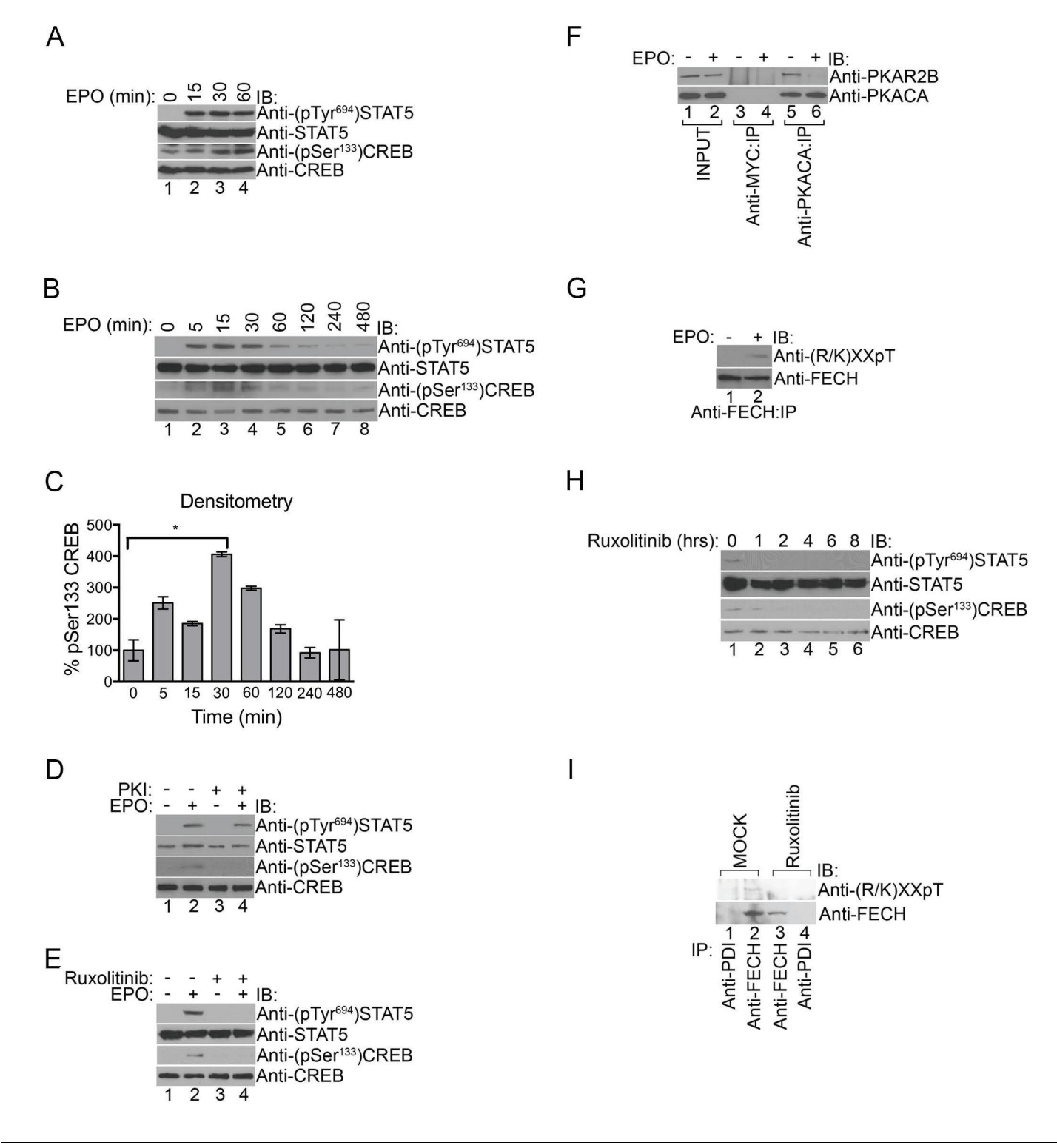

**Figure 6.** PKA links EPO signaling with heme production during erythropoiesis. (**A**) Primary murine erythroblasts were cultured from the E13.5 fetal liver, starved for 2 hr and treated with EPO (50 U/mL) for the indicated times. Western analysis showed that EPO triggered increased phosphorylation of STAT5$^{Tyr694}$ and CREB$^{Ser133}$, indicating increased JAK2 and PKA activity, respectively. (**B and C**) Human UT7 erythroid cells were serum-starved overnight, treated with EPO (2 U/mL) for the indicated times, and subjected to immunoblot analysis. EPO treatment increased both STAT5$^{Tyr694}$ and CREB$^{Ser133}$ phosphorylations similar to (**A**). A representative blot is shown in (**B**) and densitometry quantification from three independent experiments (n = 3) where the phospho-CREB signal is normalized to total CREB signal is shown in (**C**). Time points within the first 60 min were significantly different

*Figure 6 continued on next page*

*Figure 6 continued*

than time 0. (**D** and **E**) UT7 cells were treated with the indicated compounds, and immunoblot analysis was performed. CREB$^{Ser133}$ phosphorylation can be blocked by both the PKA inhibitor, 14–22, (**D**) and the JAK2 inhibitor, Ruxolitinib (**E**). In contrast, 14–22 had no effect on STAT5 phosphorylation (**D**). (**F**) UT7 cells untreated or treated with EPO were lysed and immunoprecipitated with the indicated antibodies. Western analysis showed that EPO stimulation resulted in dissociation of PKAc from PKAr. (**G**) Following PHZ treatment, primary erythroblasts were harvested from the adult murine spleen, starved for 2 hr, and stimulated with EPO (50 U/mL) for 30 min. Lysates were immunoprecipitated and subjected to immunoblot analysis that showed increased FECH phosphorylation with EPO-mediated activation. (**H** and **I**) HEL cells induced to hemoglobinize by L-ALA supplementation were treated with Ruxolitinib for 2 hr. Western blot analysis demonstrated reduced CREB$^{Ser133}$ phosphorylation (**H**) and FECH phosphorylation (**I**) with inhibition of the constitutively active JAK2 mutant. All immunoblots were performed twice unless otherwise specified. *p-value<0.05, Mean ± SEM, n = 3. IB-immunoblot; IP-immunoprecipitate; EPO-erythropoietin; PHZ-phenylhydrazine;HEL-human erythroleukemia; L-ALA–δ-aminolevulinic acid.

experiments further demonstrated that, following EPO exposure, PKAc dissociated from PKAr (*Figure 6F*), which is obligate for PKAc kinase activity (*Taylor et al., 2012*).

To determine if PKA-mediated FECH phosphorylation was regulated by EPO signaling, we performed anti-phosphothreonine immunoblot analysis on lysates from primary murine splenic erythroblasts (splenocytes) treated with either EPO or vehicle. Indeed, EPO stimulation resulted in an increase in FECH phosphorylation (*Figure 6G*). Furthermore, Ruxolitinib inhibited phosphorylation of both CREB$^{Ser133}$ and FECH$^{Thr116}$ in human erythroleukemia (HEL) cells that express the constitutively active JAK2$^{V617F}$ mutant protein (*Figure 6H and I*). Together, our data link EPO signaling to heme metabolism through PKA.

## Phosphorylated STAT5 forms a complex with PKAc

The role of PKA in erythropoiesis has remained enigmatic. This is because early studies failed to detect changes in cAMP levels despite its stimulatory effect on iron incorporation during RBC maturation that is likely mediated by the synergistic effects of CREB on STAT5 transcription (*Boer et al., 2002*, *2003*; *Gidari et al., 1971*; *Schooley and Mahlmann, 1975*). However, recent evidence indicates that PKAc activation is much more complex involving direct protein-protein interactions with other cell signaling regulators and feedback mechanisms independent of cAMP (*Taylor et al., 2012*; *Wong and Scott, 2004*; *Yang et al., 2013*, *1995*; *Zakhary et al., 2000*). Thus, we next asked whether PKAc interacts with proteins in the EPOR/JAK2/STAT5 signaling pathway, leading to its activation.

Co-immunoprecipitation experiments performed using lysates from HEL cells that have constitutive JAK2$^{V617F}$ signaling revealed that STAT5 formed a complex with PKAc (*Figure 7A*). In contrast, STAT5 could not be detected when the immunoprecipitation was performed with a control antibody (*Figure 7A*). The STAT5-PKAc complex formation was sensitive to pharmacologic inhibition of JAK2 function with Ruxolitinib (*Figure 7B*). We also tested whether EPO can trigger the formation of the STAT5-PKAc complex. STAT5 only co-precipitated with PKAc in lysates harvested from EPO-stimulated human UT7 erythroid cells (*Figure 7C*). This fraction of STAT5 was phosphorylated on Tyr694 (*Figure 7C*). In contrast, STAT5 did not co-precipitate with PKAr subunits (*Figure 7D*), suggesting that phospho-STAT5/PKAc form a distinct complex apart from PKAr. Lastly, the phospho-STAT5/PKAc complex can only be found in cell lysates derived from the cytosol but not the mitochondria (*Figure 7E*) and indicates that at least a fraction of active PKAc can diffuse freely to phosphorylate nuclear CREB (*Hagiwara et al., 1993*; *Mayr and Montminy, 2001*). Based on our results, we propose a model in which PKA signaling components are localized to the outer mitochondrial membrane during erythropoiesis by AKAP10 where it becomes activated by EPO signaling to regulate heme biosynthesis (*Figure 7F*).

## Discussion

Heme metabolism genes are downstream of GATA1 during RBC development and regulation of heme production has always been thought to occur at the level of gene transcription (*Fujiwara et al., 2009*; *Handschin et al., 2005*; *Phillips and Kushner, 2005*). However, GATA1 is required for very early stages of erythroid development when the demand for heme and hemoglobin is still low (*Fujiwara et al., 1996*). This raises the possibility that heme metabolism during cell

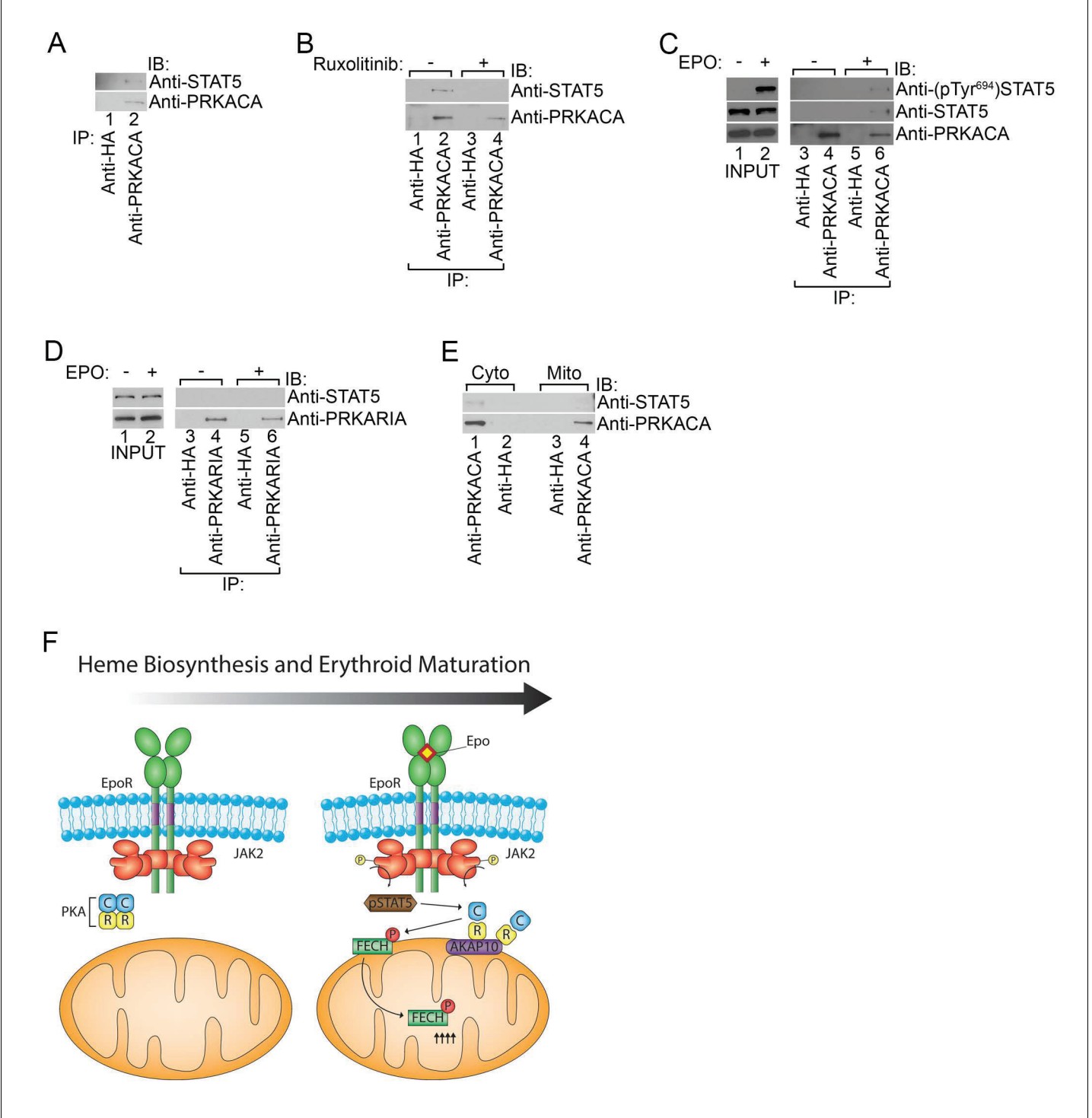

**Figure 7.** Phosphorylated STAT5 forms a molecular complex with mitochondrial PKAc. (**A**) HEL cells were lysed and immunoprecipitated with the indicated antibodies and subjected to immunoblot analysis. (**B**) HEL cells treated with MOCK or Ruxolitinib for 2 hr were subjected to similar analysis as (**A**). (**C** and **D**) UT7 cells starved overnight and treated with MOCK or EPO for 10 min were lysed and immunoprecipitated with the indicated antibodies. Immunoblot analysis was performed with the indicated antibodies. (**E**) Cytosolic and mitochondrial extracts were isolated from HEL cells and immunoprecipitated with the indicated antibodies. Bound proteins were resolved on SDS-PAGE and analyzed by immunoblotting with the indicated antibodies. (**F**) Model of how, during erythropoiesis, EPO signaling activates PKA at the mitochondrial OM that is localized by the GATA1-target, AKAP10. PKA phosphorylates FECH, which is required to achieve full FECH activity necessary to accommodate the vast heme demand for hemoglobin assembly. All immunoblots were performed twice. IB-immunoblot; IP-immunoprecipitate.

differentiation requires coordinated metabolic alterations dictated by extracellular signaling cues. EPO signaling is a critical regulator of erythropoiesis and elucidating downstream signaling pathways has been an active area of research. While it is dispensable for early erythroid specification, EPO is critical for the proliferation and survival of early erythroid progenitors (*Kuhrt and Wojchowski, 2015*) and promotes their differentiation by increasing iron uptake and reducing 'stemness' potential (*Decker, 2016*; *Ishikawa et al., 2015*; *Park et al., 2016*; *Zhu et al., 2008*). However, nothing is known regarding how EPO signaling can influence heme metabolism. Our work supports a unifying model linking the erythroid transcriptional program with a novel PKA-dependent mechanism downstream of EPO that sheds light into how heme metabolism is coupled to development.

There is a growing body of evidence that, in the absence of intrinsic apoptotic signals, mitochondrial PKA signaling within the matrix is compartmentalized due to the impermeability of the inner mitochondrial membrane (*Acin-Perez et al., 2009*; *Lefkimmiatis et al., 2013*). Under this scenario, phosphorylation of matrix proteins can only be achieved by activating signals within the matrix (*Acin-Perez et al., 2009*; *DiPilato et al., 2004*; *Lefkimmiatis et al., 2013*). However, our results support an alternative explanation in which proteins are modified prior to transport. The import of nuclear-encoded mitochondrial proteins requires the maintenance of these proteins in an unfolded state (*Lodish et al., 2012*), which would allow greater accessibility of target motifs. In support, mitochondrial membrane embedded BAX is, first, phosphorylated by OM PKA (*Danial et al., 2003*; *Harada et al., 1999*) and the requirement for protein unfolding during mitochondrial import is consistent with the low level of FECH phosphorylation *in vitro* (approximately 10%) that we observed. It is also very possible that binding of the FECH substrate to the PKA kinase induces conformational changes that would render the target motif more favorable to modification. This mechanism has previously been proposed as a means to prevent 'promiscuous' phosphorylation (*Dar et al., 2005*; *Dey et al., 2011*). Regardless the mechanism, given its exposure to the cytosol and access to proteins destined for mitochondrial localization, the OM is a prime location for such modifications to occur that would coordinate mitochondrial physiology with overall cellular behavior. Accordingly, studies have shown that OM PKA has unique signaling properties in that it is similarly responsive to cytosolic activation mechanisms but remains active much longer and is largely subject to cAMP-independent regulation (*Lefkimmiatis et al., 2013*). Our data where loss of AKAP10 and OM PKA signaling results in defective FECH modification and activity is consistent with this idea (*Figures 2* and *3*).

The regulation of PKA activity involves an intricate signaling network more complex than the canonical cAMP pathway (*Lefkimmiatis et al., 2013*; *Manni et al., 2008*; *Yang et al., 2013*). Our work implicates phosphorylated STAT5 as a novel PKAc binding protein that can displace it from autoinhibitory PKAr subunits (*Figures 6* and *7*) and is corroborated by recent work showing that phospho-STAT proteins, particularly STAT3 and STAT5, localize to mitochondria (*Carbognin et al., 2016*; *Gough et al., 2009*; *Meier and Larner, 2014*; *Wegrzyn et al., 2009*). Interestingly, in our analysis, mitochondrial expression of PKAc subunits was not as robustly increased as PRKAR2B (*Figure 1C and D*). Studies have demonstrated that, upon activation, mitochondrial OM PKAc begins to gradually diffuse throughout the cell (*Webb et al., 2008*). Thus, the non-stoichiometric increase in PKAc compared to PKAr expression is consistent with a dynamic signaling event.

To date, there have been no reported EPP-associated *FECH* mutations at the PKA target motif. However, the low prevalence of this disease has made discerning genotype-phenotype correlations difficult. Our findings that PKA is an effector of EPO/JAK2 signaling implicate PKA activity not only in the pathogenesis of EPP but also a spectrum of hematologic diseases (*Ishikawa et al., 2015*). *PRKAR1A* inactivating mutations are associated with metabolic syndromes in which anemia is prevalent (*Stratakis and Cho-Chung, 2002*). This is in agreement with murine models where *PRKAR1A* deletion has the most widespread effect and is the only knockout of the PKA family with embryonic lethality (*Stratakis and Cho-Chung, 2002*). Paradoxically, we found that it is the mitochondrial expression of PRKAR2B, and not PRKAR1A, that is most dramatically increased in maturing erythroid cells (*Figure 1C–E*). The complex nuances of PKA signaling make it very difficult to reconcile these findings. Regulatory subunits restrict both the localization as well as the activation of PKA (*Wong and Scott, 2004*), making it very challenging to distinguish the relative contributions of these two mechanisms. The dual role of PKAr is highlighted by the incomplete rescue of *PRKAR1A*$^{-/-}$-associated embryonic defects with *PRKACA* ablation (*Amieux and McKnight, 2002*). Further complicating matters is the well-documented instances of compensatory responses (*Kirschner et al., 2009*), raising the possibility that prominent roles for other PKA isoforms may simply be masked. Indeed,

there is both genetic and biochemical evidence supporting a pivotal and specific role for PRKAR2B in blood development and disease. Global transcriptome analysis has shown that *PRKAR2B* mRNA is selectively high in CD71$^+$ early erythroid cells (*Su et al., 2004*) and PRKAR2B binds with higher affinity than PRKAR1A to AKAP10 (*Burns et al., 2003*). The latter point is particularly important given the recent correlation of *AKAP10* polymorphisms with human blood traits in genome-wide association studies (*Gieger et al., 2011*). It is also notable that *AKAP10* encodes many isoforms that may localize to different subcellular compartments (*Eggers et al., 2009*; *Huang et al., 1997a*). Although our CRISPR targeting strategy was designed to specifically disrupt the N-terminal mitochondrial-targeting motif (*Figure 2G*), non-mitochondrial AKAP10 isoforms may function in a variety of contexts both in hematopoiesis and in other aspects of development including the cardiovascular system (*Kammerer et al., 2003*; *Tingley et al., 2007*). Our work, here, provides further evidence that perturbations in PKA signaling have significant impact on human health (*Kammerer et al., 2003*) including the pathogenesis of hematologic diseases that, to date, has been unappreciated and warrants further investigation.

## Materials and methods

### Cell culture

The DS19 murine erythroleukemia (MEL) subclone (RRID:CVCL_2111) was kindly provided by Arthur Skoultchi (Albert Einstein Medical College, Bronx, NY, USA). Parental human UT7 erythroid cells (RRID:CVCL_5202) were kindly provided by Meredith S. Irwin (Hospital for Sick Children, Toronto, ON, Canada). Human erythroleukemia (HEL) cells (RRID:CVCL_2481) were kindly provided by Ann Mullally (Brigham and Women's Hospital, Boston, MA, USA). All cells are mycoplasma negative and the International Cell Line Authentication Committee lists none of them as a commonly misidentified cell line. The identities of all cells were confirmed by their labs of origin since none of them are commercially available and have no standard authentication reference sample.

DS19 MEL and primary murine erythroid progenitors from E13.5 fetal liver were cultured and differentiated as previously described (*Chung et al., 2015*). Differentiating DS19 MEL cells at day 3 of 2% DMSO differentiation were treated with 10 μM (low-dose) or 50 μM (high-dose) forskolin (ThermoFisher, Waltham, MA), or 50 μM 8-Br-cAMP (Sigma-Aldrich, St. Louis, MO) for 30 min or 10 μM dmPGE$_2$ (Cayman Chemicals, Ann Arbor, MI) for 60 min and stained with *o*-dianisidine as described below. Inhibition with 20 μM H-98 (Tocris Bioscience, Minneapolis, MN), 100 nM PKI (14-22) (ThermoFisher Scientific), or 30 nM *bis*-indolylmaleimide II (Tocris Bioscience) were performed by pretreating the cells for 30 min prior to PKA pharmacologic activation.

Bulk murine erythroid progenitors from adult spleen (splenocytes) were prepared and EPO-stimulated as previously described (*Maeda et al., 2009*; *Socolovsky et al., 2001*). After two hours of rhEPO (50 U/mL) stimulation, cells were harvested and subjected to immunoprecipitation and western analysis.

Human UT7 erythroid cells were cultured in αMEM (Gibco, Gaithersburg, MD) supplemented with 20% heat-inactivated fetal bovine serum (Serum Source International, Charlotte, NC ) and 10 ng/mL GM-CSF (Peprotech, Rocky Hill, NJ). For stimulation experiments, UT7 cells were starved overnight without GM-CSF. The next morning, rhEPO (2 U/mL) was added for the indicated times prior to harvesting.

Human erythroleukemia cells (HEL) were cultured in RPMI supplemented with 10% heat-inactivated fetal bovine serum and treated with 1 μM Ruxolitinib, which was a kind gift of Dr. Ann Mullally (Brigham and Women's Hospital) and added to the cells for the indicated times.

### shRNA experiments

*Akap10*-targeting and NT9 control shRNAs in the pLKO.1-puro vector were purchased from Sigma-Aldrich. MEL cells were electroporated and monoclonal populations expressing these shRNAs were isolated as previously described (*Chung et al., 2015*). The sequences of the *Akap10* shRNAs were: shRNA-1, 5'-CCGGCCAAGTCATGTTGCGATCAATCTCGAGATTGATCGCAACATGACTTGG TTTTTG-3' and shRNA-2, 5'-CCGGGCAAGAGCACTTTAGTGAGTTCTCGAGAACTCACTAAAGTGC TCTTGCTTTTTG-3'.

## Isolation of mitochondria-enriched fractions

$1 \times 10^9$ DS19 MEL cells of each condition or $5 \times 10^8$ HEL cells were collected, washed once in cold PBS, and resuspended in 1 mL of MSHE buffer (220 mM mannitol, 70 mM sucrose, 5 mM potassium HEPES pH 7.4, 1 mM EGTA pH 7.4, supplemented with Complete EDTA-free protease inhibitor tablets [Roche, Indianapolis, IN]). Samples were then dounce homogenized and pelleted by spinning at 1000 g for 10 min at 4°C. The supernatant was separated and the leftover pellet was resuspended in 200 μL of MSHE buffer. All samples were centrifuged again at 1000 g for 10 min at 4°C. All supernatants from each undifferentiated and differentiating samples were collected and combined and centrifuged again at 1000 g for 10 min at 4°C. This supernatant was then transferred to a new tube and centrifuged at 8000 g for 20 min at 4°C. The resulting supernatant containing cytosolic proteins was transferred to another tube and flash frozen. The pellet was resuspended and washed twice more in 200 μL of MSHE buffer. The final pellet, containing mitochondrially enriched membrane fractions, was flash frozen until subjected to mass spectrometry analysis. For western analyses, the pellet was lysed in NP-40 lysis buffer. For ferrochelatase activity assays, the Mitochondrial Isolation Kit for Mammalian Cells (ThermoScientific) was used according to manufacturer's instructions. The mitochondrial pellet was resuspended in 200 μL of reaction buffer and immediately used.

## Submitochondrial fractionation and proteinase K digestion experiments

Intact mitochondria were isolated as described above without protease inhibitors and resuspended in MSHE buffer containing 5 mg/mL digitonin that was prepared fresh at 4% in water immediately before each experiment. Samples were incubated on ice for 15 min with vortexing at maximum setting every few minutes for 10 s intervals. After 15 min, samples were centrifuged at 10,000 g for 10 min at 4°C. The supernatant was transferred to a second tube and the leftover pellet was re-extracted twice more with 80 μL of MSHE containing 5 mg/mL digitonin, leaving a final pellet enriched for MPs. The supernatant from the second and third extractions were discarded. The supernatant from the first extraction was then centrifuged at 144,000 g for 1 hr at 4°C. The supernatant containing the IMS was removed and kept in a separate tube. The pellet contained the OM fraction. Proteinase K protection assays were performed as previously described (*Shirihai et al., 2000*).

## Proteomics analysis

Protein was extracted from purified mitochondria by dissolution in 8 M urea, 50 mM Tris pH 8.0, followed by probe sonication. Extracted protein was reduced and alkylated with dithiothreitol and iodoacetamide, respectively. Alkylated protein samples were first digested with endoproteinase LysC for 4 hr at ambient temperature with an enzyme to protein ratio of 200:1. Each sample was diluted with 50 mM Tris to 1.5 M urea and further digested with trypsin at an enzyme to protein ratio of 50:1, overnight and at ambient temperature. Peptides from each sample were desalted over a C18 solid phase extraction cartridge and dried down. Each sample was resuspended in 0.2 M triethylammonium bicarbonate pH 8.5, and labeled with tandem mass tag reagents (TMT) as previously described (*Hebert et al., 2013*). Labeled samples were pooled, dried, and fractionated across a strong cation exchange column (Polysulfoethyl A). Each fraction was dried, desalted, and resuspended in 0.2% formic acid.

All nano UPLC separations were performed on a nanoAcquity system. From each fraction, approximately 2 μg of peptides was injected onto a 75 μm inner diameter, 30 cm long, nano column packed with 1.7 BEH C18 particles. The mobile phases were as follows: A) 0.2% formic acid and B) 100% acetonitrile with 0.2% formic acid. Peptide were eluted with a gradient of increasing B from 0%30% over the course of 100 min, followed by a wash with 100% B and re-equilibration at 0% B. Eluting peptides were electrospray ionized and analyzed with an Orbitrap Elite mass spectrometer. The mass spectrometry analysis cycle was as follows. First a survey scan was performed with Orbitrap analysis at 60,000 resolving power at 400 *m/z*. Peptide precursors in the survey scan were sampled for ms/ms analysis by data dependent top 15 selection with dynamic exclusion turned on. Each peptide precursor selected for sampling was isolated in the ion trap, fragmented by higher energy collisional dissociation (HCD) at 35 NCE, followed by mass analysis of the fragments in the Orbitrap at resolving power 15,000 at 400 m/z.

All data analysis was performed in the COMPASS software suite (*Wenger et al., 2011*). Spectra were dssearched against a tryptic target-decoy mouse Uniprot database including protein isoforms.

Methionine oxidation and TMT on tyrosine were searched as variable modifications. Cysteine carbamidomethylation, TMT on lysine and TMT on the peptide N-terminus were searched as fixed modifications. The tolerance was set to 0.01 Da for matching fragments to the database. Matching spectra were filtered to 1% FDR at the unique peptide level based on spectral matching score (E-value) and peptide precursor ppm mass error, followed by reporter ion quantitation, protein grouping according to parsimony, and filtering to 1% FDR. Reporter ion quantitation normalization was performed essentially as previously described with the following changes (*Grimsrud et al., 2012*). First, the dataset was annotated using the MitoCarta1.0 database (*Pagliarini et al., 2008*). All proteins annotated as mitochondrial for undifferentiated and differentiated samples were averaged and linear regression was performed using Microsoft Excel. The conversion factor corresponding to the slope of the linear regression was applied to all proteins in the database regardless of MitoCarta1.0 status and the $\log_2$ fold change was calculated.

## CRISPR/Cas9-directed homology repair and *Akap10* targeting

CRISPR oligos were cloned into the px335 vector as previously described (*Chung et al., 2015*). The sequences of the CRISPR oligos were 5'-CACCGAATTTTGGGGGTTCGGCGTT-3' and 5'-AAA-CAACGCCGAACCCCCAAAATTC-3'. The single-stranded DNA oligo used to direct homology repair was 5'-TTGGAGTTTCGAAGGTGGAATAAAATCCACTCACTTATGTGTCCAATGATTTAG TAAGCTTGCACCATTCATTGCGAAGCGACGTGCGCCCAAAATTCAAGAGCAGTATCGCAGAA TCGGAGGTGGATCCCCCATCAAGATGTGGACT-3'. Targeting of *Akap10* (NM_019921.3) was performed as previously described with modifications (*Chung et al., 2015*). The CRISPR oligos were: exon-1, 5'-CTGCACTAGTCCGAAAACAG-3' and exon-3, 5'-GCAAGGCATGATTTTTAGTG-3'.

DS19 MEL cells were electroporated and cultured as previously described (*Chung et al., 2015*) along with 5 μL of 10 mM ssDNA oligo. Single clones were grown in 96-well plates and screened for the presence of the *Fech^{T115A}* allele using genomic DNA PCR followed by antisense oligonucleotide hybridization with [γ-$^{32}$P]-ATP-labeled (10 mCi/mL, specific activity = 6000 Ci/mmol, Perkin Elmer) wild-type (5'-CATCGCCAAACGCCGAACC-3') and mutant (5'-CATTGCGAAGCGACGTGCG-3') oligos as described elsewhere (*Hildick-Smith et al., 2013*). Clones harboring the mutant allele were expanded and characterized by allele specific PCR (see below) and sequenced to confirm the presence of only the mutant allele.

## High performance liquid chromatography

HPLC analysis was performed as previously described and statistical significance was determined using two-way ANOVA (*Yien et al., 2014*).

## Polymerase chain reaction and site-directed mutagenesis

Genomic DNA was isolated from DS19 MEL cells according to manufacturer's instructions (Qiagen DNeasy kit, Germantown, MD). For screening *Fech^{T115A}* knock-in clones using dot blots, the following primers were used to generate the amplicon of interest: 5'-CTGTTTGGCTCTCCTTAG-3' and 5'-GAGTCCTACTGTAACGAG-3'. For allele-specific PCRs, the latter primer from above was used with either the wild-type, 5'-CATCGCCAAACGCCGAACC-3', or mutant, 5'-CATTGCGAAGCGACG TGCG-3', forward primers. For *Akap10* genotyping, the primers used were: forward (F) primer, 5'-GAAGGGCTCGCGGACTCG-3'; reverse-1 (R1) primer, 5'-CCCTGACAAAACCCTTGC-3'; and reverse-2 (R2) primer, 5'-CACTTGCAGTGTTTTGGGGTTT-3'.

Site-directed mutagenesis was performed using the Agilent QuikChange Lightning Multi Site-Directed Mutagensis Kit (La Jolla, CA) according to the manufacturer's instructions. The following mutagenesis primers were used: T116A, 5'-CATCGCCAAACGCCGAGCCCCCAAGATTCAAG-3' and 5'-CTTGAATCTTGGGGGCTCGGCGTTTGGCGATG-3'; K113L, 5'-GCACCATTCATCGCC TTACGCCGAACCCCCAAG-3' and 5'-CTTGGGGGTTCGGCGTAAGGCGATGAATGGTGC-3'; and R114L, 5'-CCATTCATCGCCAAACTCCGAACCCCCAAGATTC-3' and 5'-GAATCTTGGGGG TTCGGAGTTTGGCGATGAATGG-3'.

## Protein purification

Recombinant His-tagged human FECH proteins were expressed and purified as previously described (*Burden et al., 1999*).

## Structural modeling

The 2.0 Å structure of human FECH (PDB 2QD4) was visualized by PyMOL (The PyMOL Molecular Graphics System, Version 1.8 Schrödinger, LLC).

## In vitro kinase assay

One microgram of His-tagged wild-type or variant human FECH was mixed with purified PKAc according to manufacturer's instructions (Promega, Madison, WI). 20% of the kinase reaction was subjected to western blot analysis. The stoichiometry of the kinase reaction was kept at 1:1. For kinase assays using [γ-$^{32}$P]-ATP (specific activity 6000 Ci/mmol, Perkin Elmer, Boston, MA), the reaction was performed with an incubation time of 30 min instead of 10 min and scaled down to where 0.2 pmol of purified FECH and PKAc were used. Four-fold excess [γ-$^{32}$P]-ATP was added. After 30 min, each reaction was immunoprecipitated with anti-FECH antibodies (see below). The incubation time for [γ-$^{32}$P]-ATP labelling was three-times longer than for other assays to ensure maximal phosphorylation in vitro. The amount of radioactivity was quantified in a scintillation counter (*Chung et al., 2015*) and normalized to immunoprecipitation efficiency, which was calculated by determining the percent of FECH protein that was recovered in the immunoprecipitation relative to input using western blotting followed by densitometry with ImageJ (*Schneider et al., 2012*). The average and standard error was calculated from three independent experiments.

## Western blotting and immunoprecipitation

All immunoblots were performed according to manufacturer's instructions and as previously described except that all phosphothreonine immunoblots were performed with an HRP-conjugated protein A secondary antibody (*Chung et al., 2015*). Anti-PDHA1 (ab67592) and anti-AKAP10 (ab97354) rabbit polyclonal antibodies were purchased from Abcam (Cambridge, MA). Mouse monoclonal anti-TUBA1A (DM1A), rabbit polyclonal anti-STAT5 (C-17), anti-PRKACB (C-20), anti-PRKAR2A (M-20), and goat polyclonal anti-FECH (C-20) and anti-HSPD1 (K-19) antibodies were purchased from Santa Cruz Biotechnology (Santa Cruz, CA). Mouse monoclonal anti-pSer133-CREB (1B6) and anti-pThr-Pro (9391S), rabbit monoclonal anti-pTyr694-STAT5 (D46E7), anti-PRKACA (D38C6), anti-SMAC (D5S4R), anti-VDAC1 (D73D12), anti-TOM20 (D8T4N), and anti-CREB (D76D11), rabbit polyclonal anti-Arg-X-X-pThr (9621S) and anti-PRKARIA (D54D9) antibodies were purchased from Cell Signaling Technology (Danvers, MA). Anti-GAPDH (MAB374) mouse monoclonal and anti-PRKARIIB (ABS14) rabbit polyclonal antibodies were purchased from Millipore (Billerica, MA). Anti-TIM23 mouse monoclonal antibody was purchased from BD Biosciences (Woburn, MA). Immunoprecipitations were performed as previously described (*Chung et al., 2010*). All immunoblots were performed two independent times except when otherwise specified. Densitometry was performed as previously described and analyzed by one-way ANOVA (*Chung et al., 2015*).

## Ferrochelatase activity assay

Ferrochelatase activity assays using purified His-tagged wild-type human FECH was performed by using 250 ng of purified protein in assay reaction buffer (0.6 M sorbitol, 40 mM HEPES pH 7.4 pH with KOH, 50 mM KCl, 1 mM MgSO$_4$). Ascorbic acid and NADH were then added to final concentrations of 0.4 mg/mL and 2 nM, respectively. The solution was incubated at 37°C for 5 min. After the incubation period, 0.2 mL of $^{55}$FeCl$_3$ (38 mCi/mL, specific activity = 54.5 mCi/mol, Perkin Elmer) was added to each sample along with either 5.5 μL of 200 μM DP or NMMP. The samples were then incubated for another 10 min at 37°C and $^{55}$Fe-radiolabeled heme was extracted as previously described and counted in a liquid scintillation counter (*Chung et al., 2015*). For activity assays using intact mitochondria, the protocol was modified to use 50 μg of freshly isolated mitochondria. All experiments were performed in duplicate and three independent experiments were performed followed by statistical analysis using student's *t*-test.

For kinetic assays, the conditions of the experiments are described elsewhere using 0.1 μM FECH, 3 μM DP, and 0.2–100 μM $^{55}$FeCl$_3$ except the reactions were carried out at 25°C (*Hunter et al., 2008*). At 30, 60, 120, 180, 300, 600, and 900 s, 10% of each reaction were removed and immediately mixed with cold FeCl$_3$ at a final concentration of 1.25 mM to stop the reaction. All samples were extracted and counted in a scintillation counter as described above. The amount of

product over time was plotted and analyzed using regression analysis with GraphPad Prism v5.0 software. Michaelis-Menten analyses with the initial velocities ($v_i$) were subsequently performed using GraphPad Prism v5.0 that calculated the maximum velocity ($v_{max}$) and Michaelis-Menten ($K_m$) constant. Experiments were performed three times and statistical analysis was performed on GraphPad Prism v5.0.

## Zebrafish experiments

Injections, *o*-dianisidine staining, and flow cytometry analysis were performed as described elsewhere (*Chung et al., 2015*). MOs were purchased from Gene Tools, LLC (Philomath, OR). Zebrafish embryos at the one-cell stage were injected with MOs targeting the exon-1/intron-1 (MO1) and exon-2/intron-2 (MO2) junctions of *D. rerio akap10* (XM_690206). The MO sequences were as follows: MO1, 5'-TGGAGCGGCCACTTCCTTACCTTTC-3'; MO3, 5'-TTTAGCACTAGACACTTACC TTTGC-3'.

## Vertebrate animal study approvals and ethics statement

All zebrafish (RRID:ZIRC_ZL1) and mouse experiments were performed in full compliance with the approved Institutional Animal Care and Use Committee (IACUC) protocols at Boston Children's Hospital (Protocol #15-07-2974R) and Brigham and Women's Hospital (Protocol #2016N000117). These studies were approved by local regulatory committees in accordance with the highest ethical standards for biomedical research involving vertebrate animals.

## Acknowledgements

We would like to thank members of our lab as well as Eva Fast, Robert I Handin, Vijay G Sankaran, and Leonard I Zon for insightful discussions and technical assistance. We thank Arthur Skoultchi (Albert Einstein Medical College, Bronx, NY, USA) for the DS19 MEL subclone and Meredith S Irwin (Hospital for Sick Children, Toronto, ON, Canada) for the human UT7 erythroid cells. We thank Leonard I Zon (Harvard Medical School, Boston, MA, USA) for the Tg(*globin-LCR*:eGFP) transgenic zebrafish line. We also thank Ann Mullally (Brigham and Women's Hospital, Boston, MA, USA) for the JAK2 inhibitor, Ruxolitinib, and the human HEL cells. We thank Eva Buy and her crew for the zebrafish animal husbandry. In vitro HPLC analyses for porphyrins were performed at the University of Utah Center for Iron and Heme Disorders (NIH U54 DK110858). This work was supported by grants from the Canadian Institutes of Health Research (CIHR Post-doctoral Fellowship, JC), the American Society of Hematology (Basic Research Fellow ASH Scholar Award, JC; Junior Faculty ASH Scholar Award, DEB), the American Cancer Society (RSG-13-379-01-LIB, TM), the Diamond Blackfan Anemia Foundation (BHP), and the National Institutes of Health (K08 DK093705, DEB; R01 DK052380, JK; R01 DK090257, JDP; P41 GM108538, JJC; R01 DK098672 and R01 GM115591, DJP; R01 DK096501, HAD; R01 DK070838, BHP; P01 HL032262, BHP, ABC, DEB, SHO, and HFL).

## Additional information

### Funding

| Funder | Grant reference number | Author |
| --- | --- | --- |
| National Heart, Lung, and Blood Institute | P01 HL032262 | Harvey F Lodish<br>Daniel E Bauer<br>Stuart H Orkin<br>Alan B Cantor<br>Barry H Paw |
| National Institute of Diabetes and Digestive and Kidney Diseases | R01 DK070838 | Barry H Paw |
| American Cancer Society | RSG-13-379-01-LIB | Takahiro Maeda |
| American Society of Hematology | | Jacky Chung<br>Daniel E Bauer |
| Canadian Institutes of Health | | Jacky Chung |

| Research | | |
| --- | --- | --- |
| National Institute of Diabetes and Digestive and Kidney Diseases | K08 DK093705 | Daniel E Bauer |
| National Institute of Diabetes and Digestive and Kidney Diseases | R01 DK052380 | Jerry Kaplan |
| National Institute of Diabetes and Digestive and Kidney Diseases | R01 DK090257 | John D Phillips |
| National Institutes of Health | R01 GM114122 | Joshua J Coon |
| National Institute of Diabetes and Digestive and Kidney Diseases | R01 DK096501 | Harry A Dailey |
| National Institutes of Health | R01 GM115591 | David J Pagliarini |
| National Institute of Diabetes and Digestive and Kidney Diseases | R01 DK098672 | David J Pagliarini |
| National Institutes of Health | P41 GM108538 | Joshua J Coon |
| National Institute of Diabetes and Digestive and Kidney Diseases | U54 DK110858 | John D Phillips |
| Diamond Blackfan Anemia Foundation | | Barry H Paw |

The funders had no role in study design, data collection and interpretation, or the decision to submit the work for publication.

### Author contributions

JC, Conceptualization, Data curation, Formal analysis, Investigation, Methodology, Writing—original draft, Writing—review and editing; JGW, CEM, ASH, LL, Data curation, Formal analysis, Investigation, Methodology; AG, Data curation, Validation, Investigation, Methodology; MM, TAD, Data curation, Investigation, Methodology; HB, Data curation, Formal analysis, Investigation; MDK, Formal analysis, Validation, Investigation, Methodology; EEC, Data curation, Formal analysis, Methodology; JK, Conceptualization, Data curation, Formal analysis, Investigation, Methodology; HFL, DEB, ABC, TM, HAD, Formal analysis, Supervision, Investigation, Methodology; SHO, Supervision, Investigation, Methodology; JDP, Data curation, Supervision, Investigation, Methodology; JJC, Formal analysis, Supervision, Validation, Investigation, Methodology; DJP, Formal analysis, Supervision, Investigation, Methodology, Project administration; BHP, Conceptualization, Resources, Supervision, Writing—review and editing

### Author ORCIDs

Johannes G Wittig, http://orcid.org/0000-0002-0598-2897
David J Pagliarini, http://orcid.org/0000-0002-0001-0087
Barry H Paw, http://orcid.org/0000-0002-0492-1419

### Ethics

Animal experimentation: In full compliance with BWH IACUC A4752-01 (Protocol #2016N000117) and BCH IACUC Protocol #15-07-2974R.

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
