## [Decision Letter]

[Editors’ note: a previous version of this study was rejected after peer review, but the authors submitted for reconsideration. The first decision letter after peer review is shown below.]

Thank you for submitting your work entitled "Erythropoietin signaling regulates heme biosynthesis." for consideration by *eLife*. Your article has been reviewed by three peer reviewers, one of whom, David Ginsburg (Reviewer #1), is a member of our Board of Reviewing Editors, and the evaluation has been overseen by Ivan Dikic as the Senior Editor. The following individuals involved in review of your submission have agreed to reveal their identity: Susan Taylor (Reviewer #3).

Our decision has been reached after consultation between the reviewers. Based on these discussions and the synthesis of all three reviews below, we regret to inform you that your work will not be considered further for publication in *eLife*.

All three reviewers agreed that this is an intriguing and well written manuscript that links regulation of heme biosynthesis to Epo activation of JAK2 and PKA signaling in the mitochondria. Specifically the authors show first using an unbiased MS analysis of mitochondrial proteins that PKA isoforms are up-regulated in the mitochondria of differentiated MEL cells. They then go on to show that ferrochetalase (FECH) has a PKA phosphorylation site and that this site is selectively phosphorylated in response to high levels of forskolin and cAMP but down regulated in the presence of PKA inhibitors. They then go on to link the increased PKA signaling to EPO stimulation of JAK2. Because the mechanism for regulation of heme biosynthesis at the metabolic level in contrast to the gene transcriptional regulation is poorly understood, this is a potentially significant advance. However, the biochemical and mechanistic evidence, including the overall elucidation of PKA signaling, is rather superficial at this point, and would require considerable additional data to justify the sweeping conclusions that are made. Specific comments/suggestions are appended at the end of this letter.

Specific comments/suggestions from the reviewers:

1) Forskolin vs. cAMP. The dose of forskolin is very high and low doses show no effect although this figure ((Figure 1—figure supplement 1) is looking at the total cell extract vs. purified mitochondria. If the authors wish to promote the idea that there are localized pools of cAMP, then the requirement for high doses of forskolin, a general cyclase activator, are counterintuitive. Isoproterenol is a much more specific activator of β adrenergic signaling and should also be tested to see if there is enhanced specificity. The difference between high vs. low levels of forskolin is disconcerting and not explained or well justified. Also it is not at all established or well accepted that high doses vs. low doses of forskolin will activate mitochondrial PKA. Mitochondrial soluble cyclase is predicted to be activated by bicarbonate, not by forskolin.

2) In the Results section the authors state "We independently confirmed these results using immunoblotting analysis where we also found increased total PKA expression in maturing erythroid cells (Figure 1 and Figure 1—figure supplement 1A). These results suggest that PKA signaling may regulate heme production". Elevation in the levels of the PKA subunits does not necessarily correlate with changes in enzyme activity. The authors may wish to modify this statement to reflect this issue.

3) It is stated in the Results that “Thr116 (humanFECH) (Figure 2). This residue is evolutionarily conserved and is present on one of the lips of the active site pocket positioned in the middle of a long α-helix". It is unusual to have a consensus phosphorylation site in the middle of a helix as this region of the secondary structure has to unravel before the phospho-transfer reaction can proceed. The authors should comment on this point.

4) Phosphorylation of FECH on the mitochondria matrix. The mitochondrial MS proteomic data shows a global enrichment of PKA signaling proteins. Specifically, the data shows an increase in RIa, RIIa, RIIb, Ca and Cb although only two of these proteins, Ca and RIIb, are actually shown in the protein blots shown in Figure 1 and the increase in the C band is actually very minimal. Although the authors state that increases in RIa and RIIa protein are also shown, in Figure 1 the levels of RIa and RIIa are not shown. The up-regulation of all of these isoforms is surprising given that there is likely to be a very specific isoform associated with PKA signaling in the matrix of the mitochondria. It is thus surprising that three of the four R-subunit isoforms are up-regulated as well as the two major C-subunit isoforms. It is extremely unlikely that all of these isoforms are in the matrix of the mitochondria as the data assumes. Perhaps several isoforms are on the outer membranes where PKA targeting is well-documented and even in the inner membrane space. But evidence supporting PKA subunits in the matrix is sparse and not independently validated in different labs. If the authors truly want to demonstrate PKA phosphorylation of FECH in the matrix of the mitochondria, they need to more rigorously validate this. Fractionation of the mitochondria could elucidate the distribution of the isoforms and show that either C or one R isoform are in the matrix. Isolated mitochondria can also be treated with cAMP to show that FECH is phosphorylated.

5) Isoform specificity. To demonstrate specificity in PKA signaling the authors need to rigorously show that one isoform, and not all three, is responsible for the phosphorylation of FECH in the mitochondria. shRNA experiments should be done to demonstrate isoform specificity. Targeting experiments can also be done. Will disruption of targeting abolish the activation of FECH? Several peptides that disrupt targeting are available. If so, can the authors discriminate between RI and RII targeting? Tools are available to carry out these experiments.

6) Mutations in PKA that lead to hematopoetic dysfunction. What are the specific mutations in PKA signaling that lead to hematopoietic dysfunction and can these also recapitulate the effects of FECH phosphorylation? These mutations will certainly also support the isoform specificity of PKA signaling.

7) Dissociation of R- and C-subunits. While high levels of cAMP can promote dissociation of the PKA subunits in cells, growing evidence suggests that unleashing of the catalytic activity of the C-subunit does not require dissociation of the R and C-subunits. These experiments with whole cells shed little light on the mechanism that is invoked here for the phosphorylation of FECH in the mitochondria. The authors show only the general dissociation of RIIb under conditions that are far from physiological. They do not address specific PKA signaling at the mitochondria, much less in the matrix of the mitochondria. PKA activity reporters are also available. Is PKA initially activated in the cytoplasm as a result of EPO binding to JAK2?

8) Total cell differences vs. mitochondrial differences. The authors need to rigorously distinguish between total cell effects vs mitochondrial effects.

9) In vitro phosphorylation of FECH. These studies carried out with recombinant expressed His-tagged FECH are also somewhat superficial. The modeling with space filling models is inadequate. The authors should at least show the individual residues and their interactions, and each residue should be clearly labeled. The phosphorylation in Figure 2 does not look robust. How long was the incubation time? Is the phosphorylation stoichiometric? Is the Km for substrate enhanced? Since recombinant proteins are available, more rigorous kinetic assays should be done, and also the authors should look to see if dimerization is influenced by phosphorylation. Is the His-tagged recombinant enzyme purified from *E. coli*? If so, there should be enough protein to do these experiments rigorously.

10) Phosphoproteomics. There should be a phosphoproteomic analysis of the mitochondrial fractionations and of the global proteins to see to what extent PKA substrates in general are upregulated.

11) PKC phosphorylation. If PKC is known to be an activator of FECH, then the authors should use this as a control to make sure that PKC cannot phosphorylate Thr116, which actually looks like a good PKC site, and also to demonstrate the robustness of activation and phosphorylation by PKC. Is the effect comparable for PKA?

12) Steps from JAK2 activation to FECH phosphorylation. There are many steps going from JAK2 activation at the plasma membrane to FECH phosphorylation in the matrix of the mitochondria, and many of these steps could be regulated by cAMP including not only phosphorylation of key metabolic enzymes such as FECH but also proteins involved in translocation from the plasma membrane to the matrix of the mitochondria. How does the signal from JAK2 activation by EPO get transmitted to the mitochondria? This mechanism is completely unclear. CREB phosphorylation at S133 is also very weak and not very convincing in the gels shown in Figure 4.

13) FECH disease mutations. What are the FECH mutations that cause EPP and do they correlate with the PKA regulation that is being proposed? Have any EPP mutations been identified at or near the FECH Thr116 phosphorylation site?

[Editors’ note: what now follows is the decision letter after the authors submitted for further consideration.]

Thank you for resubmitting your work entitled "Erythropoietin signaling regulates heme biosynthesis" for further consideration at *eLife*. Your revised article has been evaluated by Ivan Dikic (Senior editor), David Ginsberg (Reviewing editor and Reviewer #1), and two additional reviewers (Susan Taylor and John Scott).

All three reviewers agreed that the manuscript has been significantly improved. However, several key issues remain, that must be addressed satisfactorily, before acceptance for publication in *eLife* could be considered. The independent comments of all three reviewers are appended below. Though multiple comments/suggestions are included, the key issues that must be satisfactorily addressed in a revised submission are points 2, 4, 5 and 8 from reviewer #2 and comments #4 and 5 from reviewer #3.

Reviewer #1:

This revised manuscript is significantly improved and considerable new data have been added to address the initial critiques.

Reviewer #2:

Chung, et al. "Erythropoietin signaling regulates heme biosynthesis"

The authors have introduced a significant amount of new information to this revised manuscript and as a result both its significance and its impact have been greatly improved. While I would now recommend publication there are still some questions that the authors could address. Some are beyond the scope of this work but others should be answered.

1) The discovery that AKAP10 is the AKAP the mechanism by which PKA is targeted to the mitochondria is compelling and very interesting. In addition to the recent Geiger paper, we found early on that there was an advantageous mutation in AKAP10 when one compared older populations vs. younger populations (Kammerer et al., 2003). There were two mutations but the one in the AKAP binding site specifically reduced the affinity for RIa. We speculated at the time that this had a cardioprotective effect, but this could also be relevant for your findings.

2) There are some concerns about the in vitro phosphorylation in Figure 4, in particular 4E and 4F. This phosphorylation does not appear to be very robust, which is surprising. What is the stoichiometry? You should be able to do this easily with mass spec or radioactive ATP or using a gel shift assay that should distinguish the unphosphorylated protein from the phosphorylated protein.

3) In spite of the modest phosphorylated bands, the mutants look good and it is good that you did both the Thr116Ala mutant as well as the K mutants. However, did you try the phosphomimetic mutants (D/E)? This is an important experiment and should give an opposite phenotype to the Ala mutant?

4) The stoichiometry is also important for the kinetic assays. The Vmax effect seems to be rather modest and if the protein is not stoichiometrically phosphorylated then you would see a more modest effect. I suspect that the effect on activity would be even more pronounced if the protein were fully phosphorylated.

5) One last question that relates to the kinase assays that you used. What was the amount of PKA that you used relative to the amount of FECH? If it was sub-stoichiometric then you may have a situation where there is a single turnover but the substrate stays bound until some other signal releases it. If this were the case you would need to add near stoichiometric amounts of kinase and substrate to see full phosphorylation. Just a thought.

6) I do not think it is a concern that the motif is in a helix as this is a surface helix and it is likely that it is quite dynamic. I would guess that the helix propensity of this helix is not great especially because it has two prolines in it. This is unusual for a helix where Pro is typically a helix breaker. There was an earlier example of this with a structure of PRK tethered to its substrate eIG2alpha (Dar, Dever and Sicheri, 2005). In this case the helix in the substrate became disordered near the active site when the protein was tethered to the kinase. Sicheri suggests in a later paper a putative mechanism whereby the helix protects against phosphorylation until the substrate binds to the kinase.

7) Although it would not be necessary for this paper, I think that the helix could actually provide a good mechanism for docking onto the C-subunit using the groove that the PKI peptide uses. A peptide array might indicate that there were recognition motifs beyond the immediate site of phosphorylation. I would guess that the FI motif could be important for docking onto this surface. You could test this with an FI/AA mutant.

8) The pull down assays showing the C-subunit binding to Tyr phosphorylated STAT5 is intriguing but still preliminary. The mechanism is unclear. Is the RIIb subunit no longer associated with this complex? ie Does the binding of STAT5 induce the dissociation of the holoenzyme or does it just cause a conformational change that opens up the active site making is accessible to FECH? Does the entire complex pull down so that it is poised for binding to FECH? Are additional PKA substrates phosphorylated other than FECH or are you looking at the events in a specific signaling complex? Is the C-subunit now released into the cytoplasm and no longer tethered to the mitochondria or is it still tethered to the mitochondria? Do you trap a complex with STAT5 and FECH if you use the Ala mutant of FECH? As you do not fully dissociate the complex even in the presence of cAMP, which is consistent with the very early Johnson results, which you reference, and also with earlier mass spec data and Scott's recent papers. The high forskolin experiments would force the holoenzyme apart but this is not physiological.

9) One last point. AKAP10 has a PDZ motif at its C-terminus that immediately follow the A Kinase Binding motif. In kidney this interacts with PDZK1 but I do not know if other binding partners have been discovered. It could just be something to look for in your various data sets.

In conclusion, this is a very nice story and the biological and physiological relevance is obviously very high. If some of the above questions can be satisfactorily addressed, publication is recommended.

Reviewer #3:

This is an extensively revised version of a manuscript that was previously submitted to *eLife* on how erythroid transcriptional programming regulates post-translational mechanisms that influence heme metabolism. The quality of the western blot characterization of the phenomena in Figure 2 and Figure 4 remains a weakness of the study. The most substantive change in the manuscript is the inclusion of some new and potentially interesting data on the role of AKAP10 as a mitochondrial adapter for the location of PKA subunits. I have several issues pertaining to this new data.

1) AKAP10 has two locations that could be pertinent to this work. It is found at the plasma membrane where it is believed to have some RGS function and on or in the mitochondria. The membrane-associated fraction of AKAP10 may play an important role and needs to be considered by the authors. A few comments in the discussion would suffice.

2) In subsection “Mitochondrial PKA is localized to the outer mitochondrial membrane via AKAP10” it states that AKAP10 has long been recognized as a downstream target for the GATA1 erythroid lineage transcription factor. This is not immediately apparent in the Fujiwara et al., 2009 paper (text or supplementary figures). Please clarify.

3) On the basis of other studies it is quite possible that deletion of AKAP10 results in the up-regulation of other mitochondrial anchoring proteins. This is an easily performed set of experiments. Also, the authors should incorporate analysis of WAVE1 they cite the paper that identified it at the mitochondria but never follow up on this.

4) The experiments in Figure 3 are still marred by the lack of analyses using physiological agonists of the cAMP signaling pathway. Contrary to what is stated in paragraph two of subsection “Mitochondrial outer membrane PKA signaling regulates hemoglobinization and erythropoiesis”. Forskolin is not a PKA agonist. Forskolin is a supraphysiolgical activator of adenylyl cyclases. Consequently, other cAMP responsive factors such as Epac and CNG channels will be mobilized. This issue was raised by the reviewers before and has still not be satisfactorily addressed.

5) The data linking PKA and Stat5 seems rather weak. The authors need to deftly describe how biochemical effects at the mitochondria with PKA relate to changes in nuclear CREB phosphorylation. Especially since Cam Kinases, Mist1 & Mist 2 and Akt are just as potent modulators of Ser 133 on CREB.

[Editors' note: further revisions were requested prior to acceptance, as described below.]

Thank you for resubmitting your work entitled "Erythropoietin signaling regulates heme biosynthesis" for further consideration at *eLife*. Your revised article has been favorably evaluated by Ivan Dikic (Senior editor), a Reviewing editor, and two reviewers.

The manuscript has been improved but there are some remaining issues that need to be addressed before acceptance, as outlined below:

1) All reviewers were concerned about the quality of Figure 4, which should be replaced. Also, discussion should be included as to why the level of phosphorylation is so low, as well as an explanation of what efforts were made to increase the level of phosphoryaltion.

2) Given that the sample is only 10% phosphorylated, interpretation of the kinetic data in Figure 5 should be specifically addressed.

3) If possible, the manuscript would be strengthened by including a native gel for the in vitro phosphorylated protein, looking for a gel shift to confirm the quantification reported. Also, would it be possible to separate the phospho protein by ion exchange chromatography?

---

## [Author Response]

[Editors’ note: the author responses to the first round of peer review follow.]

*All three reviewers agreed that this is an intriguing and well written manuscript that links regulation of heme biosynthesis to Epo activation of JAK2 and PKA signaling in the mitochondria. Specifically the authors show first using an unbiased MS analysis of mitochondrial proteins that PKA isoforms are up-regulated in the mitochondria of differentiated MEL cells. They then go on to show that ferrochetalase (FECH) has a PKA phosphorylation site and that this site is selectively phosphorylated in response to high levels of forskolin and cAMP but down regulated in the presence of PKA inhibitors. They then go on to link the increased PKA signaling to EPO stimulation of JAK2. Because the mechanism for regulation of heme biosynthesis at the metabolic level in contrast to the gene transcriptional regulation is poorly understood, this is a potentially significant advance. However, the biochemical and mechanistic evidence, including the overall elucidation of PKA signaling, is rather superficial at this point, and would require considerable additional data to justify the sweeping conclusions that are made. Specific comments/suggestions are appended at the end of this letter.*

On behalf of all the authors, I want to thank you for your efforts in evaluating our work. We are very encouraged by the positive tone of the reviews and the in-depth analysis provided by everyone is reflective of the broad interest and provocative nature our work. Your insightful feedback has propelled us to explore new concepts. Based upon our new data, we have revised our original conclusions and developed a new unifying model where transcriptional and post-translational mechanisms work in concert to regulate heme metabolism. These new additional results have significantly strengthened our revised manuscript, which is now composed of 7 main figures instead of 4 in the original submission. In particular, our new results include:

1) in vivo evidence using an animal model that this EPO-PKA signaling cascade is important in a physiologic setting.

2) Biochemical and genetic evidence that FECH phosphorylation occurs on the outer mitochondrial membrane and not in the mitochondrial matrix.

3) A mechanism of how PKA become localized, which links the erythroid transcriptional program with EPO signaling and also explains our observed PKA- isoform-specific effects.

4) Extensive quantitative analysis including kinetic measurements of enzymatic activity of phosphorylated FECH.

5) Direct evidence of how erythropoietin signaling regulates PKA activity.

Our manuscript uncovers a novel signal transduction pathway that couples cell differentiation with heme metabolism – two fundamental cellular processes that were, previously, thought to occur independently. All reviewers agreed that our work is of significant interest and addresses this fundamental biological problem that, to date, has been unresolved. We strongly believe that our revised manuscript would strongly appeal to the broad readership of *eLife*, and respectfully, ask that the Editorial Office and the reviewers give our revised work consideration.

*Specific comments/suggestions from the reviewers:*

*1) Forskolin vs. cAMP. The dose of forskolin is very high and low doses show no effect although this figure ((Figure 1—figure supplement 1) is looking at the total cell extract vs. purified mitochondria. If the authors wish to promote the idea that there are localized pools of cAMP, then the requirement for high doses of forskolin, a general cyclase activator, are counterintuitive. Isoproterenol is a much more specific activator of β adrenergic signaling and should also be tested to see if there is enhanced specificity. The difference between high vs. low levels of forskolin is disconcerting and not explained or well justified. Also it is not at all established or well accepted that high doses vs. low doses of forskolin will activate mitochondrial PKA. Mitochondrial soluble cyclase is predicted to be activated by bicarbonate, not by forskolin.*

Indeed, the contribution of distinct pools of cAMP and their corresponding activating signals are two significant controversies in the field. They are interrelated issues because the roles of various intracellular PKA pools have typically been extrapolated from pharmacologic studies. We thank the reviewer for bringing these key points to our attention. Over the past several months, we have worked hard to experimentally address these issues and we would like the opportunity to discuss how they address the reviewer’s queries.

A significant proportion of our new results involve the identification and characterization of AKAP10 as a mitochondrial scaffold that nucleates PKA at the mitochondrial outer membrane. Much of what we know of AKAP10 is from studies carried out by Dr. Susan Taylor (reviewer #3) over the past 20 years (Huang et al., 1997; Wang et al., 2001; and Eggers, Schafer, Goldenring and Taylor, 2009). Recently, AKAP10 has garnered a great deal of attention since its association with human cardiac function and hematologic parameters in genome-wide association studies (GWAS) (Gieger et al., 2011)

Our new work demonstrates that the majority of mitochondrial PKA in maturing red cells is actually tethered onto the outer mitochondrial membrane, which is dependent upon AKAP10. In the absence of AKAP10, we found a significant depletion of PKA in this compartment and reduced FECH phosphorylation and function (Figure 2, Figure 4, and 5). Inhibition of AKAP10 in vivo also results in anemia (Figure 3). This suggests that modification of FECH occurs prior to its import into the mitochondrial matrix where it has enzymatic function.

As it specifically relates to the Reviewer’s comments, we would like to begin by clarifying that our conclusion is not that cAMP is the sole activator of PKA during erythropoiesis. In fact, as we note later (see point #12 below and Figure 7 in our revised manuscript), we believe that PKA activation can be achieved in several ways. Thus, while we use pharmacologic manipulation of PKA activity many of which are cAMP agonists, we only do so in an effort to examine the relevance of the PKA pathway in principle.

When it comes to understanding the role of any compartment of PKA, we also strongly believe that our combined biochemical and genetic data offer new insight compared to previous work by other groups that rely upon extrapolation from pharmacologic studies for two reasons. First, dose-dependent and cell-type-specific effects have been documented (Humphries, Pennypacker and Taylor, 2007 and Lefkimmiatis, Leronni and Hofer, 2013), suggesting that discrepancies between independent studies may reflect technical differences. Second, the methods in which suborganellular PKA activity is monitored has been debated, making it even more difficult to reconcile differences that are observed between studies (DiPilato, Cheng and Zhang, 2004 and Acin-Perez, et al., 2009). Thus, not only do our new data uncover a novel physiologic pathway, it also sheds light into this controversy from a different scientific vantage point.

The reviewer also astutely noted that we use total cell extract instead of purified mitochondria to perform our assays, implying that the latter approach would be superior. In this case, we, respectfully, disagree. The use of purified mitochondria to study signaling pathways overlooks the dynamic nature of signal transduction. As previous studies have demonstrated, proteins (including PKA substrates, such as BAX) can be modified prior to mitochondrial transport (Harada et al., 1999 and Danial et al., 2003). Furthermore, mitochondrial outer membrane PKA is a compartment with distinct biochemical properties that can be activated by cytosolic PKA agonists such as forskolin while having vastly different kinetics (Lefkimmiatis, Leronni and Hofer, 2013). Our characterization of the role of AKAP10 is actually consistent with previous work in which case, treatment of purified mitochondria with PKA pharmacologic agents would only lead to ambiguous negative results.

In addition to including in vivo genetic and biochemical data on the role of AKAP10 into our revised manuscript, we have also included a much more thorough discussion of these topics. We are very excited that our work has elicited such a fantastic discussion, which underscores our belief that *eLife* is an excellent venue to publish our revised manuscript as it deserves a broad audience. We hope that the reviewer is satisfied with our responses.

*2) In the Results section the authors state "We independently confirmed these results using immunoblotting analysis where we also found increased total PKA expression in maturing erythroid cells (Figure 1 and Figure 1—figure supplement 1A). These results suggest that PKA signaling may regulate heme production". Elevation in the levels of the PKA subunits does not necessarily correlate with changes in enzyme activity. The authors may wish to modify this statement to reflect this issue.*

We apologize to the reviewer for our misleading statement. We have now revised this section to limit our conclusions derived from this figure only to PKA expression. This sentence now reads, “We independently confirmed these results using immunoblotting with isoform-specific antibodies where we also found increased total expression of these PKA subunits in maturing erythroid cells (Figure 1).”

*3) It is stated in the Results that “Thr116 (humanFECH) (Figure 2). This residue is evolutionarily conserved and is present on one of the lips of the active site pocket positioned in the middle of a long α-helix". It is unusual to have a consensus phosphorylation site in the middle of a helix as this region of the secondary structure has to unravel before the phospho-transfer reaction can proceed. The authors should comment on this point.*

This is an excellent point raised by the reviewer. It is true that access to the consensus motif is crucial for kinase phosphorylation. Our new results demonstrating the involvement of AKAP10 at the outer mitochondrial membrane corroborates this paradigm since the import of proteins across this membrane requires that proteins remain in an unfolded state (Lodish et al., 2012). This would increase accessibility of this site to PKA. Notably, this mechanism of regulation by PKA has been observed previously (Harada et al., 1999 and Danial et al., 2003). Outer membrane PKA phosphorylates BAX prior to its integration within the mitochondrial membrane (Harada et al., 1999 and Danial et al., 2003). These points are of significant interest to basic cell signaling and we have now included a discussion of this in our revised manuscript.

*4) Phosphorylation of FECH on the mitochondria matrix. The mitochondrial MS proteomic data shows a global enrichment of PKA signaling proteins. Specifically, the data shows an increase in RIa, RIIa, RIIb, Ca and Cb although only two of these proteins, Ca and RIIb, are actually shown in the protein blots shown in Figure 1 and the increase in the C band is actually very minimal. Although the authors state that increases in RIa and RIIa protein are also shown, in Figure 1 the levels of RIa and RIIa are not shown. The up-regulation of all of these isoforms is surprising given that there is likely to be a very specific isoform associated with PKA signaling in the matrix of the mitochondria. It is thus surprising that three of the four R-subunit isoforms are up-regulated as well as the two major C-subunit isoforms. It is extremely unlikely that all of these isoforms are in the matrix of the mitochondria as the data assumes. Perhaps several isoforms are on the outer membranes where PKA targeting is well-documented and even in the inner membrane space. But evidence supporting PKA subunits in the matrix is sparse and not independently validated in different labs. If the authors truly want to demonstrate PKA phosphorylation of FECH in the matrix of the mitochondria, they need to more rigorously validate this. Fractionation of the mitochondria could elucidate the distribution of the isoforms and show that either C or one R isoform are in the matrix. Isolated mitochondria can also be treated with cAMP to show that FECH is phosphorylated.*

In our revised manuscript, we have addressed these concerns. As the reviewer has noted, PKA refers to an enzyme complex consisting of regulatory and catalytic subunits. In humans, there are three catalytic subunits (a, b, and g) and four regulatory subunits (RIa, RIIa, RIb, and RIIb) (Kirschner, Yin, Jones and Mahoney, 2009). Mice do not have PKA-catalytic-g (Kirschner, Yin, Jones and Mahoney, 2009) and, thus, was not detected in our analysis using a murine model. In addition, the expression of RIa is restricted to neurons (Kirschner, Yin, Jones and Mahoney, 2009) and is similarly not detected in our analysis. We apologize to the reviewer for omitting these critical details, which we have now included in our revised manuscript. We have also performed a more thorough expression analysis of all PKA isoforms and included it in Figure 1 including submitochondrial localization (Figure 2).

The points brought-up by the reviewer made us realize that we were premature in our conclusions. We performed a significant amount of new experiments that now demonstrate that FECH is phosphorylated by PKA at the outer mitochondrial membrane. To avoid being repetitive, we kindly ask the reviewer refer to our responses to points #1 and #3 regarding our new data and their implications on the PKA signaling.

*5) Isoform specificity. To demonstrate specificity in PKA signaling the authors need to rigorously show that one isoform, and not all three, is responsible for the phosphorylation of FECH in the mitochondria. shRNA experiments should be done to demonstrate isoform specificity. Targeting experiments can also be done. Will disruption of targeting abolish the activation of FECH? Several peptides that disrupt targeting are available. If so, can the authors discriminate between RI and RII targeting? Tools are available to carry out these experiments.*

We thank the reviewer for bringing up this point. Isoform specificity is very difficult to address in PKA signaling because of functional redundancy and compensatory responses both of which have been well documented (Kirschner, Yin, Jones and Mahoney, 2009 and Amieux and McKnight 1997). In fact, in mouse knockouts of individual PKA regulatory subunits compensation of the other isoforms is frequently found (Cummings DE, et al., Nature, 1996; Kirschner, Yin, Jones and Mahoney, 2009 and Amieux and McKnight, 1997). In addition, loss-of-function shRNA experiments do not adequately discriminate between total and mitochondrial PKA effects as noted by the reviewer in point #8 below. Thus, we, respectfully, do not believe that the approach recommended by the reviewer will yield satisfactory and unambiguous scientific answers.

We agree that aberrant targeting of PKA away from mitochondria is critical to deciphering a mitochondrial role for PKA. However, instead of using peptides, we have uncovered an adaptor protein – AKAP10 – that is a transcriptional target of GATA-1, the master regulatory erythroid transcription factor (Huang, et al., 1997 and Wang et al., 2001). Strikingly, inhibition of AKAP10 in erythroid cells as well as a zebrafish model resulted in reduced heme production and anemia (Figure 3). Moreover, as the reviewer suggested, we also examined how mislocalization of PKA away from mitochondria via AKAP10 inhibition affected FECH phosphorylation and activity. We found that loss of AKAP10 led to significantly diminished FECH phosphorylation and FECH activity (Figure 4 and Figure 5).

In our revised manuscript, we have included a much more thorough discussion on the interplay of the various PKA isoforms as well as included our analysis of the role of AKAP10 as a dual-specificity AKAP in heme biosynthesis. We sincerely hope that we have satisfactorily addressed the reviewer’s concerns.

*6) Mutations in PKA that lead to hematopoetic dysfunction. What are the specific mutations in PKA signaling that lead to hematopoietic dysfunction and can these also recapitulate the effects of FECH phosphorylation? These mutations will certainly also support the isoform specificity of PKA signaling.*

We thank the reviewer for asking these two critical questions. As the reviewer has noted, point #6 is very much related to point #5 above and, to avoid being repetitive, we kindly ask the reviewer to refer to point #5 for our response to isoform specificity.

The most prevalent PKA mutations associated with hematologic symptoms are deletions of PRKAR1A (also known as PKA-regulatory 1 α subunit). This is not to say that other isoforms do not have a role. We are aware that different PKA isoforms have different biochemical properties (for example, Cheng X, et al., J Biol Chem, 2001). However, as we noted in our response to point #5, compensatory responses has made it very difficult to address isoform specificity using classical genetic loss-of-function approaches in vivo. In addition, as noted in reviewer point #8 below, global inhibition of PKA isoforms (with RNAi or CRISPR), would be highly ineffective at delineating cytosolic and mitochondrial effects.

In contrast, our characterization of AKAP10 addresses the issue previously raised by the reviewer in an elegant and biologically relevant fashion. AKAP10-dependent recruitment of PKA requires the regulatory subunits (Wong W and Scott JD, Nat Rev Mol Cell Biol, 2004). Thus, loss of AKAP10 essentially mimics loss of PKAr specifically from mitochondria while leaving all other PKA signaling components intact. Strikingly, we found that AKAP10 deletion in erythroid cells resulted in diminished FECH phosphorylation and activity. Interestingly, recent GWAS studies have found that AKAP10 variants are associated with hematologic blood traits (Grieger C, et al., Nature, 2011). We have included all these details in the expanded Discussion section of our revised manuscript.

*7) Dissociation of R- and C-subunits. While high levels of cAMP can promote dissociation of the PKA subunits in cells, growing evidence suggests that unleashing of the catalytic activity of the C-subunit does not require dissociation of the R and C-subunits. These experiments with whole cells shed little light on the mechanism that is invoked here for the phosphorylation of FECH in the mitochondria. The authors show only the general dissociation of RIIb under conditions that are far from physiological. They do not address specific PKA signaling at the mitochondria, much less in the matrix of the mitochondria. PKA activity reporters are also available. Is PKA initially activated in the cytoplasm as a result of EPO binding to JAK2?*

We thank for the reviewer for raising these concerns, which has propelled us to explore new possibilities to explain our observations. Our new experiments demonstrate that activated PKA responsible for phosphorylating FECH is *not* in the mitochondrial matrix (please see points #1, #5, and #6).

With regards to how EPO signaling activates PKA, our new results argue that PKA is activated directly through phosphorylated STAT5 in a cAMP-independent fashion. As we have discussed, early studies in red blood cells failed to detect changes in cAMP levels throughout red cell maturation (Boer, Drayer and Vellenga, 2003; Gidari, Zanjani and Gordon, 1971; and Schooley and Mahlmann, 1975). This is not to say that PKA has no role in erythropoiesis but rather it may be cAMP-independent. In support, exogenous cAMP promotes iron uptake in red cells by phospho-CREB dependent synergism with STAT5 (Boer, Drayer and Vellenga, 2002). Thus, there is clearly cross talk between these two pathways that does not involve endogenous cAMP.

In our new Figure 7, we show using human erythroid cells, that phosphorylated STAT5 forms a complex with PKA-catalytic subunits. This supports a model whereby EPO signaling leads to phosphorylation of STAT5 that can displace and activate PKA- catalytic subunits. Our new model also explains why CREB – the proto-typical non- mitochondrial PKA target – is phosphorylated by EPO signaling. This is because the fraction of PKA that is activated in response to EPO is *not* matrix. Rather, it is cytosolic, which also activates mitochondrial outer membrane PKA as previously shown by other groups (Lefkimmiatis, Leronni and Hofer, 2013).

We appreciate the enthusiasm demonstrated by the reviewer in our work. However, we believe that unraveling the kinetics behind PKA activation and whether dissociation between the regulatory and catalytic subunits is absolutely required in red cells is beyond the scope of the current manuscript. The goal of our work here is not to test every nuance of PKA signaling that has been previously published; follow-up studies are more appropriate to address all these issues. Instead, the key conceptual advance is that PKA is activated by EPO and heme metabolism is regulated by this mechanism. Lastly, we would like to emphasize that, akin to EPO signaling, iron uptake induced by cAMP does not result in iron overload (Boer, Drayer and Vellenga, 2003; Gidari, Zanjani and Gordon, 1971; and Schooley and Mahlmann, 1975). This is a key point because it suggests that PKA activation can not only trigger iron import but also coordinate iron assimilation into heme. Our new data, model, and in-depth discussion are now included in our revised work.

*8) Total cell differences vs. mitochondrial differences. The authors need to rigorously distinguish between total cell effects vs mitochondrial effects.*

Please see our responses to points #5 and #6.

*9)* In vitro *phosphorylation of FECH. These studies carried out with recombinant expressed His-tagged FECH are also somewhat superficial. The modeling with space filling models is inadequate. The authors should at least show the individual residues and their interactions, and each residue should be clearly labeled. The phosphorylation in Figure 2 does not look robust. How long was the incubation time? Is the phosphorylation stoichiometric? Is the Km for substrate enhanced? Since recombinant proteins are available, more rigorous kinetic assays should be done, and also the authors should look to see if dimerization is influenced by phosphorylation. Is the His-tagged recombinant enzyme purified from E. coli? If so, there should be enough protein to do these experiments rigorously.*

We thank the reviewer for noting these technical issues and we apologize for not providing sufficient detail. We have now performed extensive in vitro analysis to examine enzyme kinetics (Figure 5 and Figure 5—figure supplement 1). In addition, we have repeated many of our immunoblots. Many of our early experiments were performed using a secondary antibody that resulted in high background signals. Recently, we switched to a protein-A sepharose HRP-conjugate that is much more effective at reducing background signals. We have now included these new experiments in Figure 4.

*10) Phosphoproteomics. There should be a phosphoproteomic analysis of the mitochondrial fractionations and of the global proteins to see to what extent PKA substrates in general are upregulated.*

We absolutely agree with the reviewer that further confirmation of PKA activation would greatly strengthen our work. This is why we are excited to report that we have addressed this in two ways. Our response to this reviewer concern consists of unpublished and ongoing work and we would kindly request that our response be held in strict confidence and not be released for public knowledge.

First, using phospho-antibody-specific flow cytometry analysis, a colleague (Dr. Vijay Sankaran, Children’s Hospital Boston, Harvard Medical School) has independently found that PKA substrates are hyper-phosphorylated in response to EPO treatment of CD34+ human hematopoietic progenitor cells. This was presented at the recent 58th American Society of Hematology Annual Meeting in San Diego, CA (Kim AR, et al., 58th ASH Annual Meeting, San Diego, CA and https://ash.confex.com/ash/2016/webprogram/Paper93991.html). They are in the process of publishing their work and, thus, while we are excited by an independent validation of our model, we are also eager to publish our own work to capitalize on its novelty.

Second, given that Dr. Sankaran has already validated hyper-phosphorylation of PKA substrates, we focused our efforts on understanding the functional relevance of CREB phosphorylation. We addressed this by using bioinformatics to determine if a “CREB- gene signature” can be found within the erythroid transcriptional program. We isolated previously identified genes that were direct targets of CREB (Impey, et al., Cell, 2004) and superimposed it onto gene expression databases that have been well annotated according to stages of erythroid differentiation (Zhang, Socolovsky, Gross and Lodish, 2003). In Figure 8 is a box-and-whisker (Tukey method, GraphPad Prism) plot of our data showing changes in the expression of genes as a function of the stages of R1 to R5 stages of erythroid development. White bars represent changes in the expression of all 16,384 genes that were detected by Zhang, Scolovsky, Gross and Lodish and the red bars represent CREB target genes annotated by Impey, et al.

Author response image 1.**DOI:**
http://dx.doi.org/10.7554/eLife.24767.016

At every stage of erythroid development, we found that this “CREB gene signature” was indeed up-regulated in red blood cells (*p* < 0.05). We are in the process of performing careful analysis on the relevance of the PKA/CREB transcriptional program in erythropoiesis the results of which are beyond the scope of the current manuscript. However, taken together, the biochemical analysis in our revised manuscript along with the analysis performed by Dr. Sankaran in human CD34+ cells and the preliminary gene expression bioinformatics analysis, we believe that we have made a strong and convincing argument that PKA is important in red blood cell development.

*11) PKC phosphorylation. If PKC is known to be an activator of FECH, then the authors should use this as a control to make sure that PKC cannot phosphorylate Thr116, which actually looks like a good PKC site, and also to demonstrate the robustness of activation and phosphorylation by PKC. Is the effect comparable for PKA?*

We thank the reviewer for suggesting using PKC as a specificity control. We are excited to say that we have now performed the suggested experiments (Figure 3—figure supplement 1). PKC inhibition using *bis*-indolylmaleimide II (Mahata M, et al., Mol Pharmacol, 2002) was unable to block PKA-dependent heme production suggesting that they function in two independent and parallel pathways. In addition, we have performed additional experiments using pharmacologic PKA agents on erythroid cells harboring the Thr115Ala FECH mutation (Figure 5). When we treat these mutant erythroid cells with forskolin, we failed to detect any changes in hemoglobinization suggesting that the Thr115 site is specific for PKA activity.

*12) Steps from JAK2 activation to FECH phosphorylation. There are many steps going from JAK2 activation at the plasma membrane to FECH phosphorylation in the matrix of the mitochondria, and many of these steps could be regulated by cAMP including not only phosphorylation of key metabolic enzymes such as FECH but also proteins involved in translocation from the plasma membrane to the matrix of the mitochondria. How does the signal from JAK2 activation by EPO get transmitted to the mitochondria? This mechanism is completely unclear. CREB phosphorylation at S133 is also very weak and not very convincing in the gels shown in Figure 4.*

We thank the reviewer for their interest in this mechanism. We agree that it is a major question and have addressed it in our revised manuscript. To avoid being repetitive, we kindly request that the reviewer visit our response to point #7 in which we go into great detail explaining our new data.

With regards to our blots in Figure 4, we would like to first assure the reviewer that all of our immunoblots were performed at least twice. We apologize for the poor quality of some of our experiments and have addressed this concern in three independent ways. First, we have explicitly stated the replicate number for all immunoblots in the Figure Legends and the Materials and methods section. Second, for the changes in CREB phosphorylation, we have performed a third independent experiment and quantified our results using densitometry with statistical analysis (it is now new Figure 6). Finally, we have repeated many of these experiments using reagents that further minimize background signals (please see our response to point #9). Our Materials and methods section now reflect these technical changes.

*13) FECH disease mutations. What are the FECH mutations that cause EPP and do they correlate with the PKA regulation that is being proposed? Have any EPP mutations been identified at or near the FECH Thr116 phosphorylation site?*

We thank the reviewer for asking these astute questions and have now included an appropriate discussion in our manuscript. In short, mutation of the phosphorylation site has yet been documented in EPP. However, this does not indicate that this pathway is not important in this disease for a number of reasons.

First, EPP is a rare disorder and because of its low prevalence, one of the major hurdles in understanding the genetics of rare diseases is finding a comprehensive genotype- phenotype correlation. For example, it took 11 years from the initial reports of *GATA1* (Tsai SF, et al., Nature, 1989 and Evans T and Felsenfeld G, Cell, 1989) to the first identification of humans with *GATA1* mutations (Nichols KE, et al., Nat Genet, 2000). To date, the only *FECH* mutation that is consistently found in EPP is a hypomorphic allele that is inherited in combination with other mutations (Gouya L, et al., Nat Genet, 2002 and Whatley SD, et al., Br J Dermatol, 2010). The search for these other genetic lesions is an active area of study. In fact, the characterization of an X-linked, dominant EPP-like disease that is associated with C-terminal mutations in *ALAS2* highlights the emerging molecular and genetic heterogeneity of EPP (Whatley SD, et al., Am J Hum Genet, 2008 and Balwani M, et al., Mol Med, 2013).

Second, complete loss of FECH function has never been found in EPP and one very likely explanation is that severe FECH deficiencies are non-viable (Whatley SD, et al., Br J Dermatol, 2010). This would further complicate any attempts at finding genotype- phenotype correlations since *FECH* mutations must only occur in certain combinations in order for EPP to develop.

We are actively engaged in efforts to collect more clinical data from EPP patients to see if the FECH Thr116 site or the surrounding motif is mutated in EPP. In addition, we are also examining whether regulators of the EPO/PKA pathway can account for the 4% of EPP cases in which the *FECH* gene is not mutated (Whatley SD, et al., Br J Dermatol, 2010). We have included an appropriate discussion covering this topic in our revised work and hope that the reviewer finds it satisfactory.

[Editors' note: the author responses to the re-review follow.]

*Reviewer #2:*

*Chung, et al. "Erythropoietin signaling regulates heme biosynthesis"*

*The authors have introduced a significant amount of new information to this revised manuscript and as a result both its significance and its impact have been greatly improved. While I would now recommend publication there are still some questions that the authors could address. Some are beyond the scope of this work but others should be answered.*

*1) The discovery that AKAP10 is the AKAP the mechanism by which PKA is targeted to the mitochondria is compelling and very interesting. In addition to the recent Geiger paper, we found early on that there was an advantageous mutation in AKAP10 when one compared older populations vs. younger populations (Kammerer et al,. 2003). There were two mutations but the one in the AKAP binding site specifically reduced the affinity for RIa. We speculated at the time that this had a cardioprotective effect, but this could also be relevant for your findings.*

We thank the reviewer for bringing this study to our attention and apologize for having missed it throughout the course of preparing our manuscript. It would be interesting to see if AKAP10 mutations in older populations correlate with hematologic parameters. We are also actively searching GWAS databases to determine if there is a link between AKAP10 mutations and porphyrias (EPP in particular). We have included the Kammerer et al., 2003 reference into our discussion.

*2) There are some concerns about the* in vitro *phosphorylation in Figure 4, in particular 4E and 4F. This phosphorylation does not appear to be very robust, which is surprising. What is the stoichiometry? You should be able to do this easily with mass spec or radioactive ATP or using a gel shift assay that should distinguish the unphosphorylated protein from the phosphorylated protein.*

We thank the reviewer for suggesting this experiment. Indeed, we have used [γ-32P]-ATP labeling and determined that in vitro, the amount of phosphorylated FECH is 9.75 ± 3.18%. This result has now been included in the main text with an accompanying description of the protocol in the Materials and methods section.

*3) In spite of the modest phosphorylated bands, the mutants look good and it is good that you did both the Thr116Ala mutant as well as the K mutants. However, did you try the phosphomimetic mutants (D/E)? This is an important experiment and should give an opposite phenotype to the Ala mutant?*

This is an excellent suggestion. We have actually tried this experiment with the phosphomimetic mutant using in vitropurified proteins. Unfortunately, we were unable to detect any significant change in enzyme activity. This has lead us to hypothesize that the phospho-group somehow interacts with the porphyrin/iron substrates in a manner that we still do not fully understand and will require a great deal of additional structural biology experiments to decipher, which are beyond the time line for the revision.

*4) The stoichiometry is also important for the kinetic assays. The Vmax effect seems to be rather modest and if the protein is not stoichiometrically phosphorylated then you would see a more modest effect. I suspect that the effect on activity would be even more pronounced if the protein were fully phosphorylated.*

We agree that stoichiometry is an important consideration when interpreting the results of our kinetic assays. As we discussed in our responses to points 2 and 3, not all FECH protein is phosphorylated in vitroand, as a complementary approach, we have tried to examine the activity of His-tagged phospho-mimetic mutant FECH proteins without much success. Future work is needed to better understand the structural changes that occur with FECH upon modification that would provide important clues as to the relationship between stoichiometry and effect on catalysis.

*5) One last question that relates to the kinase assays that you used. What was the amount of PKA that you used relative to the amount of FECH? If it was sub-stoichiometric then you may have a situation where there is a single turnover but the substrate stays bound until some other signal releases it. If this were the case you would need to add near stoichiometric amounts of kinase and substrate to see full phosphorylation. Just a thought.*

For our in vitroassays, the ratio of PKA to FECH is approximately 1:1. We had never considered that PKA may remain bound to FECH following the modification and thank the reviewer for bringing this to our attention. We have also included a small note regarding the stoichiometry of the in vitrokinase reaction in our Materials and methods section as well as the corresponding Figure Legends.

*6) I do not think it is a concern that the motif is in a helix as this is a surface helix and it is likely that it is quite dynamic. I would guess that the helix propensity of this helix is not great especially because it has two prolines in it. This is unusual for a helix where Pro is typically a helix breaker. There was an earlier example of this with a structure of PRK tethered to its substrate eIG2alpha (Dar, Dever and Sicheri, 2005). In this case the helix in the substrate became disordered near the active site when the protein was tethered to the kinase. Sicheri suggests in a later paper a putative mechanism whereby the helix protects against phosphorylation until the substrate binds to the kinase.*

We thank the reviewer for the insightful feedback and providing us with their expertise. Indeed, we have considered that this phosphorylation event may be highly dynamic, which would explain its location within a helix. Furthermore, we have also wondered how stable this helical structure would remain throughout enzyme catalysis (ie. “helix propensity” as referred to by the reviewer). We have included these details and accompanying references (Dar, Dever and Sicheri, 2005 and Dey et al., 2011) into our discussion to further strengthen our manuscript.

*7) Although it would not be necessary for this paper, I think that the helix could actually provide a good mechanism for docking onto the C-subunit using the groove that the PKI peptide uses. A peptide array might indicate that there were recognition motifs beyond the immediate site of phosphorylation. I would guess that the FI motif could be important for docking onto this surface. You could test this with an FI/AA mutant.*

We thank the reviewer for their interest in our work. The role of PKA signaling in hematopoiesis is an emerging concept (our work and also North et al., 2007; Kim, PG, et al., J Exp Med, 2015; and Jing L, et al., J Exp Med, 2015), and we are actively exploring avenues to pursue for follow-up studies.

*8) The pull down assays showing the C-subunit binding to Tyr phosphorylated STAT5 is intriguing but still preliminary. The mechanism is unclear. Is the RIIb subunit no longer associated with this complex? ie Does the binding of STAT5 induce the dissociation of the holoenzyme or does it just cause a conformational change that opens up the active site making is accessible to FECH? Does the entire complex pull down so that it is poised for binding to FECH? Are additional PKA substrates phosphorylated other than FECH or are you looking at the events in a specific signaling complex? Is the C-subunit now released into the cytoplasm and no longer tethered to the mitochondria or is it still tethered to the mitochondria? Do you trap a complex with STAT5 and FECH if you use the Ala mutant of FECH? As you do not fully dissociate the complex even in the presence of cAMP, which is consistent with the very early Johnson results, which you reference, and also with earlier mass spec data and Scott's recent papers. The high forskolin experiments would force the holoenzyme apart but this is not physiological.*

We thank the reviewer for their enthusiasm and are excited to announce that we have performed additional experiments in an effort to further address the reviewer’s questions. Some of these concerns are also related to concerns raised by reviewer 3 point 5.

First, we performed additional co-immunoprecipitation experiments using antibodies directed against PKAr subunits. The goal was to examine whether phosphorylated STAT5 formed a macromolecular complex with PKAc and PKAr. In contrast to Figure 7 to Figure 7, STAT5 did not co-precipitate with PKAr (new Figure 7). As it relates to the reviewer’s queries, our result suggests that the phospho-STAT5/PKAc complex is distinct from that of PKAr. It would also suggest that phospho-STAT5 may cause the PKAc/PKAr complex to dissociate. We have considered ways to further explore this latter point. However, it is technically very challenging. We do not believe that using traditional STAT5 truncation mutants would be informative since disruption of its interaction with the EPOR/JAK2 complex would similarly block activation of PKA (please see Figure 6 showing that JAK2 activity is required for PKA activation). It is also possible that the phospho-STAT5/PKAc complex is indirect further limiting our options regarding in vitrobinding assays with bacterially purified proteins.

The second experiment we performed is to use mitochondrial lysates for coimmunoprecipitations. This would help address the issue of whether the phospho-STAT5/PKAc complex is tethered to the mitochondria. Our results show that we could not detect a stable complex between phospho-STAT5 and PKAc using isolated mitochondria (new Figure 7). This result is consistent with a model where AKAP10 recruits PKA to the outer mitochondrial membrane (OMM) via PKAr. Upon EPO activation, STAT5 becomes phosphorylated and phospho-STAT5 dislodges PKAc from PKAr at the OMM. This model would also explain how CREB is also phosphorylated in response to EPO since the activation of PKA is not strictly restricted to mitochondria.

Whether dissociation of the PKAr/PKAc complex is absolutely required for kinase activity is a difficult issue to address in our system. We were never able to completely disrupt the PKAr/PKAc complex with EPO stimulation (see Figure 6). Thus, it is possible that the kinase is active even as a macromolecular complex. We hope that our latest experiments have adequately addressed the reviewer’s comments.

*9) One last point. AKAP10 has a PDZ motif at its C-terminus that immediately follow the A Kinase Binding motif. In kidney this interacts with PDZK1 but I do not know if other binding partners have been discovered. It could just be something to look for in your various data sets.*

We thank the reviewer for their enthusiasm for our proteomics screen. We are in the process of validating additional hits that would be of biological significance.

*Reviewer #3:*

*This is an extensively revised version of a manuscript that was previously submitted to eLife on how erythroid transcriptional programming regulates post-translational mechanisms that influence heme metabolism. The quality of the western blot characterization of the phenomena in Figure 2 and Figure 4 remains a weakness of the study. The most substantive change in the manuscript is the inclusion of some new and potentially interesting data on the role of AKAP10 as a mitochondrial adapter for the location of PKA subunits. I have several issues pertaining to this new data.*

*1) AKAP10 has two locations that could be pertinent to this work. It is found at the plasma membrane where it is believed to have some RGS function and on or in the mitochondria. The membrane-associated fraction of AKAP10 may play an important role and needs to be considered by the authors. A few comments in the discussion would suffice.*

We thank the reviewer for this comment that raises an important issue regarding isoform specificity. When AKAP10 was first identified by Huang, LJ, et al., PNAS, 1997, they found multiple isoforms with different molecular weights. These isoforms may possess different N-terminal signal peptides which would direct AKAP10 to various subcellular compartments. This is a main reason why we designed our CRISPR deletion strategy to disrupt the N-terminal mitochondrial-targeting motif – to specifically address the mitochondrial role of AKAP10 in the event that multiple isoforms were expressed. Certainly, we were unable to discriminate between various isoforms in our zebrafish knockdown experiments. We have also considered the possibility that human AKAP10 mutations may have non-mitochondrial effects that would account for the observed cardiovascular phenotype (please see point 1 by reviewer #2). To address these reviewer concerns, we have, now, included a discussion in the main text of our revised manuscript as well as the corresponding Figure Legend to explicitly state that our CRISPR deletion strategy eliminates the N-terminal motif.

*2) In subsection “Mitochondrial PKA is localized to the outer mitochondrial membrane via AKAP10” it states that AKAP10 has long been recognized as a downstream target for the GATA1 erythroid lineage transcription factor. This is not immediately apparent in the Fujiwara et al. 2009 Mol Cell paper (text or supplementary figures). Please clarify.*

We apologize for the confusion. *AKAP10* mRNA was found to be upregulated by Fujiwara, T, et al., Mol Cell, 2009 in Supplementary Table 3. There were two probe sets for AKAP10 – one found a significant induction while the other did not. Based on this, and the lack of an obvious GATA-1 binding site in the immediately upstream AKAP10 promoter, very little follow-up work was performed. We would also like to note that expression profiling of maturing erythroid cells independently performed by Zhang, Socolovsky, Gross and Lodish, 2003, also found a similar induction of AKAP10 transcript as murine red blood cells differentiated (please see raw RNA-seq data in Supplementary files in Zhang, Socolovsky, gross and LodishJ, 2003). We have now included this second reference to further support our results. In addition, we have also re-phrased that sentence to emphasize that these previous reports were high-throughput studies that require detailed analysis of raw data sets.

*3) On the basis of other studies it is quite possible that deletion of AKAP10 results in the up-regulation of other mitochondrial anchoring proteins. This is an easily performed set of experiments. Also, the authors should incorporate analysis of WAVE1 they cite the paper that identified it at the mitochondria but never follow up on this.*

We thank the reviewer for bringing this to our attention since it has important implications on our future work. Over the past two years, we have tried exhaustively to use CRISPR genome editing to delete *akap10* in zebrafish. To date, we have failed to get germline transmission of a null allele despite evidence of a deletion event in the parental generation. As the reviewer noted, we hypothesize that there are compensatory responses (such as upregulation of other AKAPs), in murine models not found in zebrafish. While our efforts, so far, have been met with disappointment, we believe that it may actually pave the way for the use of zebrafish to study AKAP mechanisms since adaptive responses may be different than currently used cell culture and murine model systems. Our analysis is only preliminary and beyond the scope of the current work.

*4) The experiments in Figure 3 are still marred by the lack of analyses using physiological agonists of the cAMP signaling pathway. Contrary to what is stated in paragraph two of subsection “Mitochondrial outer membrane PKA signaling regulates hemoglobinization and erythropoiesis”. Forskolin is not a PKA agonist. Forskolin is a supraphysiolgical activator of adenylyl cyclases. Consequently, other cAMP responsive factors such as Epac and CNG channels will be mobilized. This issue was raised by the reviewers before and has still not be satisfactorily addressed.*

We apologize for our oversight. We have now performed additional sets of experiments to address this issue (Figure 3). It has been previously shown that prostaglandin E2 (PGE2) is a physiologic PKA agonist that promotes several aspects of hemopoiesis (North et al., 2007 and Goessling et al., 2009). We treated our erythroid cells with dimethyl-PGE2 (dmPGE2), which is a more stable analog of PGE2 and found that it also promoted hemoglobinization similar to other PKA activating agents that we have tested (Figure 3). The effects of dmPGE2 were inhibited by PKI (14-22) (Figure 3), providing more physiologic evidence and specificity that PKA modulation is a determinant of heme production during red cell maturation.

*5) The data linking PKA and Stat5 seems rather weak. The authors need to deftly describe how biochemical effects at the mitochondria with PKA relate to changes in nuclear CREB phosphorylation. Especially since Cam Kinases, Mist1 & Mist 2 and Akt are just as potent modulators of Ser 133 on CREB.*

This is a similar concern raised by reviewer 2 point 8. In our work, we use Ser133 CREB phosphorylation as a simply marker for PKA activity. We understand that other kinases have been reported to also phosphorylate the same site and our work with the PKA inhibitor, 14-22 (Figure 6), provides strong evidence that in our work, Ser133 CREB phosphorylation results from elevated PKA activity.

As the reviewer has noted, previous work suggests that CREB is phosphorylated in the nucleus after passive diffusion of active PKAc into the nucleus (Hagiwara et al., 1993 and reviewed by Mayr and Montminy, 2001). This suggests that unlike the mitochondrial matrix that appears to have a distinct pool of PKA (Acin-Perez et al., 2009 and Lefkimmiatis, Leronni and HoferK, 2013), the interplay between cytosolic PKA activation and nuclear CREB responses is much more indiscriminate. Our new data shows that the phospho- STAT5/PKAc complex can only be detected in the cytosolic, non-mitochondrial compartment (Figure 7), indicating that some cytosolic PKAc becomes activated that can trigger nuclear CREB phosphorylation. We hope that our new data and accompanying brief discussion (found in the Figure 7 Results section) meets the reviewers standards.

[Editors' note: further revisions were requested prior to acceptance, as described below.]

*The manuscript has been improved but there are some remaining issues that need to be addressed before acceptance, as outlined below:*

*1) All reviewers were concerned about the quality of Figure 4, which should be replaced. Also, discussion should be included as to why the level of phosphorylation is so low, as well as an explanation of what efforts were made to increase the level of phosphoryaltion.*

We have now elaborated on an explanation for the low level of phosphorylation in vitroin our Discussion section. We increased incubation time from 10 minutes to 30 minutes for [γ-32P]-ATP labeling experiments to help reach maximal phosphorylation and have now included a note regarding this procedural modification in both the main text as well as the “Materials and methods” section.

We have also repeated the experiment for Figure 4 and replaced it with one that is more acceptable. We hope the reviewers are satisfied with the improved figure quality.

Overall, we have independently repeated these results several times and using complimentary approaches. We strongly believe that we have provided compelling evidence that FECH is a PKA target.

*2) Given that the sample is only 10% phosphorylated, interpretation of the kinetic data in Figure 5 should be specifically addressed.*

All enzyme kinetic constants are a function of temperature. Our kinetic assays were performed at room temperature while single time-point experiments were carried out at 37oC. This is a major caveat that limits the interpretation of in vitroresults. It is not technically feasible to perform a radioactive labeling assay at the desired 30-second intervals at 37oC. We have included a brief note regarding this caveat in our revised work (Results section).

Although the extent of Fech phosphorylation at the non-physiological room temperature was ~10%, this apparent modest reaction is at the terminus of a multi-step signalling cascade starting with Epo stimulation. Therefore, one would expect an “amplification effect” which would greatly magnify the apparent small changes in direct phosphorylation. Examples of such amplification effect would be the coagulation cascade or the complement system.

*3) If possible, the manuscript would be strengthened by including a native gel for the* in vitro *phosphorylated protein, looking for a gel shift to confirm the quantification reported. Also, would it be possible to separate the phospho protein by ion exchange chromatography?*

We thank the reviewers for their interest in our work. Actually, we are trying to develop a system where native phosphoproteins are produced by engineered *E. coli* (Park et al., 2011, Science and Lajoie et al., 2014, Science). If we are successful, we will be able to not only perform more detailed enzymatic assays but also obtain more structural data regarding how phosphorylation may influence enzymatic activity. These efforts would require more in-depth analysis that are, however, beyond the scope of the current manuscript and beyond the reasonable, allocated time frame for a revision.